# Striking a Balance: An Optimal Mechanism Design for Heterogenous Differentially Private Data Acquisition for Logistic Regression

## Abstract

We investigate the problem of solving ML tasks from data collected from privacy-sensitive sellers. Since the data is private, sellers must be incentivized through payments to provide their data. Thus, the goal is to design a mechanism that optimizes a weighted combination of test loss, seller privacy, and payment, i.e., strikes a balance between getting a good privacy-preserving ML model and limiting payments to the sellers. To do this, we first solve logistic regression with known heterogeneous differential privacy guarantees. We then consider the main problem where the differential privacy requirements are decided by the buyer to balance the tradeoff between test loss and payments. To solve this problem, we use our earlier result on logistic regression with known privacy guarantees along with standard mechanism design theory to formulate an optimization problem which is nonconvex. We establish conditions under which the problem can be convexified using a change of variables technique. This insight is then harnessed to develop an algorithm that provides optimal solution. Additionally, we demonstrate the resilience of our mechanism to scenarios in which data points and privacy sensitivities are correlated. Finally, we demonstrate the utility of our algorithm by applying it to the Wisconsin breast cancer dataset.

## 1 Introduction

Machine learning (ML) applications have experienced significant growth in recent years. Furthermore, substantial efforts have been dedicated to ensuring the privacy of training data, with the prevalent adoption of differential privacy. While existing literature presents various algorithms to guarantee differential privacy, a lingering question persists: determining the optimal degree of differential privacy. For instance, opting for a higher level of differential privacy may compromise the performance of the machine learning model, yet it significantly enhances privacy protection for the data provider. Therefore, along with considering the model performance through metrics such as misclassification loss, we need to also consider the privacy loss of the data providers (also referred by sellers). In this paper, we delve into addressing this nuanced tradeoff by formulating a mechanism which balances competing objectives: achieving a high quality ML model while minimizing the privacy loss experienced by data providers.

To further motivate our problem, in practice, while some of the data for training ML models are publicly available, sensitive information such as health or financial data may not be shared due to privacy concerns. For instance, a hospital may want to use health vitals to predict heart disease, but patients may be reluctant to share such data due to privacy concerns. Moreover, each patient would have a different cost for the same loss of privacy (which we term privacy sensitivity). Addressing this, there is a growing interest in encouraging data sharing through two strategies: (i) introducing noise to ML model's weights to enhance dataset anonymity, and (ii) providing compensation to data sellers to offset potential privacy risks Posner & Weyl (2019), Kushmaro (2021). The amount of compensation that is provided to patients would, of course, depend on their privacy sensitivity and their privacy loss, which can be measured using differential privacy. To operationalize this concept and thus accurately represent the tradeoff between model performance and privacy loss, we propose a robust mathematical framework for designing a data market. This market

facilitates data acquisition from privacy-conscious sellers, utilizing (a) mechanism design to incentivize sellers to truthfully disclose their privacy sensitivity (we consider that sellers can lie about their privacy sensitivities) and (b) statistical learning theory to strike a balance between payments and model accuracy. The market involves a buyer seeking data from privacy-sensitive sellers, aiming to construct a high-quality ML model while minimizing overall payments to sellers. Conversely, individual sellers seek fair compensation for potential privacy compromises. Therefore, through the data market, we capture the tradeoff between designing a good ML model while ensuring that privacy loss to data providers is small.

We are motivated by the work in Fallah et al. (2023), which considers mean estimation of a scalar random variable using data from privacy-sensitive sellers. Our objective here is to design a mechanism for the more challenging and practically useful problem of logistic regression with vector-valued data. Now, to consider the tradeoff between payments and model accuracy, it is imperative to mathematically represent the buyer's objective, i.e., model accuracy. In contrast to Fallah et al. (2023), where the buyer's objective simplifies to the variance of a mean estimator, which they assume to be known, our scenario considers the buyer's objective to be the expected misclassification error of a logistic regression model in which the statistics of the dataset are unknown. To address this issue, we propose using Rademacher complexity to model the buyer's objective. Furthermore, most prior work on differentially private logistic regression, such as (Chaudhuri et al. (2011), Ding et al. (2017)) consider homogeneous differential privacy, in which every individual has the same privacy guarantees. However, our approach acknowledges the practical reality that sellers might have different degrees of willingness (privacy sensitivity) to share their data. Therefore, we consider that each data point has to be privacy protected differently (which is done by considering heterogeneous differential privacy), leading to different utility of each data point in contributing to the ML model.

To summarize, our goal is to design a mechanism for the buyer to optimize an objective that trades off between classification loss and payments to sellers while also taking into account the differential privacy requirements of the sellers. Our contributions are as follows:

- In section 3, we provide an approach to accurately model the misclassification loss for logistic regression. We further highlight that unlike the case of the same differential privacy for all users as in Chaudhuri et al. (2011), our objective for logistic regression should include an additional regularization term for achieving optimal test loss performance.

- Next, we build upon the above result to solve our mechanism design problem (section 4). For this problem, we provide a payment identity that determines payments as a function of the differential privacy guarantees. This is used to design an objective for the mechanism design problem. Further, we show that the objective can be made convex through a change in variables trick for a large class of model parameters. Subsequently, we propose an algorithm to optimally solve the mechanism design problem. We note that, in practice, if we consider health data, certain segments of society may have poorer health outcomes than other segments and it is possible that those segments of the society may be less sensitive to privacy considerations. In other words, it is possible that the data and privacy sensitivities are correlated. Our model allows for such correlations.

- We also perform an asymptotic analysis by considering large number of sellers to understand how much it will cost a buyer to obtain sufficient data to ensure a certain misclassification loss in the ML model that results from the mechanism. The interesting insight here is that, because the buyer can selectively choose sellers to acquire data from, the budget required for a given bound on the misclassification loss is bounded.

- Finally, we demonstrate the application of our proposed mechanism on the Wisconsin breast cancer data set UCI (1995). We observe fast convergence, indicating the usefulness of the change of variables.

## 1.1 Related Work

**Differentially Private ML Algorithms:** While literature on creating differentially private data markets is relatively sparse, there is a vast literature on incorporating differential privacy in statistical modeling and learning tasks. For example, Cummings et al. (2015) builds a linear estimator using data points so that

there is a discrete set of privacy levels for each data point. Nissim et al. (2012), Ghosh et al. (2014), Nissim et al. (2014), and Ligett et al. (2017) use differential privacy to quantify loss that sellers incur when sharing their data. McMahan et al. (2017) demonstrates training of large recurrent language models with seller-level differential privacy guarantees. Works such as (Alaggan et al. (2017), Nissim et al. (2014), Wang et al. (2015), Liao et al. (2020), Ding et al. (2017)) also consider problems concerned with ensuring differential privacy. Some works consider a different definition of privacy. Roth & Schoenebeck (2012), Chen et al. (2018), and Chen & Zheng (2019) use a menu of probability-price pairs to tune privacy loss and payments to sellers. Perote-Peña & Perote (2003), Dekel et al. (2010), Meir et al. (2012), Ghosh et al. (2014), Cai et al. (2014) consider that sellers can submit false data. In the context of differentially private ML algorithms, a portion of our work can be viewed as contributing to differentially private logistic regression with heterogeneous sellers.

**Mechanism Design:** Mechanism design has a long history, originally in economics and more recently in algorithmic game theory. Recent work such as Abernethy et al. (2019) considers auctions in which buyers bid multiple times. Chen et al. (2018) provides a mechanism that considers minimizing worst-case error of an unbiased estimator while ensuring that the cost of buying data from sellers is small. However, in this paper, the cost is chosen from a discrete set of values. Other papers, such as Ghosh & Roth (2015), Liu & Chen (2017), and Immorlica et al. (2021), also consider mechanism design for different objectives and problems of interest. However, none of these works incorporates ML algorithms or differential privacy in their analysis. As mentioned earlier, our work is more closely related to Fallah et al. (2023), where the authors develop a mechanism to estimate the mean of a scalar random variable by collecting data from privacy-sensitive sellers. However, unlike Fallah et al. (2023) where they assume some statistical knowledge of the quantity to be estimated, our challenge is to design a mechanism without such knowledge. This leads to interesting problems in both deriving a bound for the misclassification loss and solving a non-convex optimization problem to implement the mechanism.

## 2 Logistic Regression while Ensuring Heterogeneous Differential Privacy

For designing the optimal mechanism, a major challenge is to represent the misclassification error of the logistic regression problem. We first formally define differential privacy and then introduce the problem of repesenting the misclassification error.

### 2.1 Differential Privacy

To build the necessary foundation, we define the notion of privacy loss that we adopt in this paper. We assume that sellers trust the platform to add necessary noise to the model weights to keep their data private. This is called central differential privacy. The first definition of differential privacy was introduced by Dwork et al. (2006) which considered heterogenous differential privacy (same privacy guarantee to all data providers). In our paper, we consider a slight extension wherein all the users are provided different privacy guarantees, i.e., heterogeneous differential privacy. More formally, it is defined as follows.

**Definition 1 [Alaggan et al. (2017)]:** Let $\boldsymbol{\epsilon} = (\epsilon_i)_{i=1}^m \in \mathbb{R}_+^m$. Also, consider $S$, $S'$ be two datasets that differ in $i^{th}$ component with $|S|$ being cardinality of set $S$. Let $\mathbb{A}$ be an algorithm that takes a dataset as input and provides a vector in $\mathbb{R}^n$ as output. We say that the algorithm provides $\boldsymbol{\epsilon}$-centrally differential privacy, if for any set $V \subset \mathbb{R}^n$,

$$e^{-\epsilon_i} \leq \frac{\mathbb{P}[\mathbb{A}(S) \in V]}{\mathbb{P}[\mathbb{A}(S') \in V]} \leq e^{\epsilon_i} \quad \forall i \in \{1, 2, \ldots, |S|\}. \tag{1}$$

This definition states that if the value of $\epsilon_i$ is small, then it is difficult to distinguish between the outputs of the algorithm when the data of seller $i$ is changed. Note that a smaller value of $\epsilon_i$ means a higher privacy guarantee for the seller.

### 2.2 Representing the Misclassification Error

In this subsection, we formulate and solve the problem of representing the misclassification error in logistic regression with heterogeneous user privacy requirements. To do this, we initially focus on a related yet simpler

scenario: logistic regression with heterogeneous differential privacy requirements. Later, when we consider the mechanism, the results in this subsection will be used.

In this section, we consider the following problem:

- We have a set of $m$ users, with user $i$ having data point $z^i = (\boldsymbol{x}^i, y^i)$, where we assume $\|\boldsymbol{x}^i\| \leq 1 \;\; \forall i$. We let $D = \{(\boldsymbol{x}^1, y^1), \ldots, (\boldsymbol{x}^m, y^m)\}$ denote the data set.

- Each user $i$ demands that $\epsilon_i$ differential privacy must be ensured for their data.

- The platform aims to design best estimator $\boldsymbol{w}$ by minimizing misclassification loss $\mathbb{E}[\mathbb{I}_{\{sign(\boldsymbol{w}^T\boldsymbol{x}) \neq y\}}]$ such that $\|\boldsymbol{w}\| \leq \beta$ while ensuring differential privacy $\boldsymbol{\epsilon}$.[1]

Chaudhuri et al. (2011) present an algorithm to solve logistic regression while ensuring homogeneous differential privacy, i.e., each user demands same differential privacy. A natural extension of their algorithm for minimizing heterogenous differential privacy can be stated as follows (proof that this algorithm ensures heterogeneous differential privacy is presented in Proposition 1 of the appendix):

**Algorithm 0:**

1. Choose $(\boldsymbol{a}, \eta) \in \mathbb{F}$, where the constraint set is $\mathbb{F} = \{(\boldsymbol{a}, \eta) : \eta > 0, \sum_i a_i = 1, a_i > 0, a_i \leq k/m, a_i\eta \leq \epsilon_i \; \forall i\}$ where $k$ is a fixed constant.

2. Pick $\boldsymbol{b}'$ from the density function $h(\boldsymbol{b}') \propto e^{-\frac{\eta}{2}\|\boldsymbol{b}'\|}$. To ensure this, we pick $\|\boldsymbol{b}'\| \sim \Gamma(n, \frac{2}{\eta})$ and direction of $\boldsymbol{b}'$ uniformly at random. Let $\boldsymbol{b}' = \frac{2\boldsymbol{b}}{\eta}$. Thus, $\|\boldsymbol{b}\| \sim \Gamma(n, 1)$ and direction chosen uniformly at random.

3. Given a dataset $D$ and differential privacies $\boldsymbol{\epsilon}$, compute $\hat{\boldsymbol{w}} = \text{argmin}_{\boldsymbol{w}}\hat{\mathbb{L}}(D, \boldsymbol{w}, \boldsymbol{a}, \eta)$, where $\hat{\mathbb{L}}(D, \boldsymbol{w}, \boldsymbol{a}, \eta) = \sum_{i=1}^{m} a_i \log(1 + e^{-y^i \cdot \boldsymbol{w}^T\boldsymbol{x}^i}) + \boldsymbol{b}'^T\boldsymbol{w} + \frac{\lambda}{2}\|\boldsymbol{w}\|^2$ for some $\lambda > 0$. Output $\hat{\boldsymbol{w}}$.

Proposition 1 shows that any choice of $(\boldsymbol{a}, \eta)$ satisfying the aforementioned constraints satisfies $(\epsilon_i + \Delta(m, \lambda))$-differential privacy requirements for $\Delta(m, \lambda) = 2\log\left(1 + \frac{k}{m\lambda}\right)$.

**Remark**: Note that in most machine learning models, $m$ is large enough such that $m\lambda >> 1$ and thus the term $\Delta$ is much smaller than the differential privacy guarantees used in practice. Therefore, for brewity, we consider $(\epsilon_i + \Delta(m, \lambda)) \approx \epsilon_i$ for further analysis.

However, it is unclear how to choose $\boldsymbol{a}$ and $\eta$ to get a good test error performance. For example, one can choose $a_i = 1/m$ and $\eta = m\min_i \epsilon_i$. However, such a choice is clearly unable to exploit the fact that we need to protect some data points more than others. Besides, our numerical experiments in Section 5 demonstrate that solving $\min_{\boldsymbol{w}}\hat{\mathbb{L}}(D, \boldsymbol{w}, \boldsymbol{a}, \eta)$ over feasible set of $(\boldsymbol{a}, \eta)$ does not provide good results. Therefore, to understand how to choose $(\boldsymbol{a}, \eta)$, we appeal to statistical learning theory to get an upper bound on $\mathbb{E}[\mathbb{I}_{\{sign(\boldsymbol{w}^T\boldsymbol{x}) \neq y\}}]$ in terms of $\hat{\mathbb{L}}(D, \boldsymbol{w}, \boldsymbol{a}, \eta)$ with the true loss. This leads to following result.

**Theorem 2.1.** *Given a classification task, let $D$ be the dataset from $m$ users with $\|\boldsymbol{x}^i\| \leq 1$. Further, consider that users have differential privacy requirements $\boldsymbol{\epsilon} = (\epsilon_i)_{i=1}^m \in \mathbb{R}_+^m$ respectively. Also, let $\mathbb{L}(D, \boldsymbol{c}; \boldsymbol{\epsilon}, \boldsymbol{w}) = \mathbb{E}[\mathbb{I}_{\{sign(\boldsymbol{w}^T\boldsymbol{x}) \neq y\}}]$ be misclassification loss and $\hat{\mathbb{L}}(D, \boldsymbol{w}, \boldsymbol{a}, \eta)$ be as defined above. Then, the following holds for appropriate $\mu, \sigma$ with probability at least $(1 - \delta)(1 - \delta')$ for every choice of $\boldsymbol{\epsilon} \in \mathbb{R}^m$ and $(\boldsymbol{a}, \eta) \in \mathbb{F}$ and $\boldsymbol{w}$ chosen such that $\|\boldsymbol{w}\| \leq \beta$ for some $\beta > 0$.*

$$\mathbb{E}[\mathbb{I}_{\{sign(\boldsymbol{w}^T\boldsymbol{x}) \neq y\}}]| \leq \hat{\mathbb{L}}(D, \boldsymbol{w}, \boldsymbol{a}, \eta) + \mu(\delta, \beta)\|a\| + \sigma(\delta, \delta', \beta)\left(\frac{1}{\eta}\right). \tag{2}$$

---

[1] $\|\boldsymbol{w}\|$ is the $l_2$ norm of $\boldsymbol{w}$, and $\mathbb{I}_{\{\cdot\}}$ is the indicator function.

Using the above bound for generalization error, we add additional regularization terms, namely $\mu||\boldsymbol{a}||$ and $\sigma/\eta$ to incorporate generalization loss in the objective function.

$$\min_{\boldsymbol{a},\eta,\boldsymbol{w}}\left[\sum_{i=1}^{m}a_i\log(1+e^{-y^i\cdot\boldsymbol{w}^T\boldsymbol{x}^i})+\frac{\lambda}{2}||\boldsymbol{w}||^2+\frac{2\boldsymbol{b}^T\boldsymbol{w}}{\eta}+\mu||\boldsymbol{a}||+\frac{\sigma}{\eta}\right]$$

$$\text{s.t.}(\boldsymbol{a},\eta)\in\mathbb{F}\text{ and }\mu,\sigma,\lambda\text{ are hyperparameters} \tag{3}$$

Further, we can choose an appropriate value of $\lambda$ to satisfy the constraint $||\boldsymbol{w}||\leq\beta$. Therefore, the objective in Eq. (3) with the mentioned constraints can be used to solve logistic regression with heterogeneous differential privacy requirements. Additionally, Eq. (3) serves as a proxy for representing misclassification loss. In section 4, we will discuss algorithmic considerations in solving the above optimization problem and also provide an algorithm to optimally choose parameters $(\mu,\sigma,\lambda)$ using validation data.

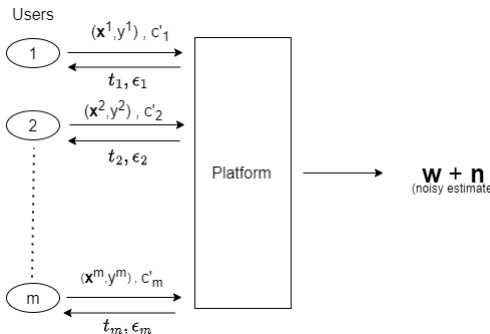

Figure 1: Interaction between sellers and the platform

## 3    Mechanism Design

We will now use our logistic regression result to consider the mechanism design problem. We consider a platform (buyer) interested in collecting data from privacy-sensitive users (sellers) to build a logistic regression model. Further, sellers may have different costs associated with the privacy lost by sharing their data, i.e., they may have different privacy sensitivities. Therefore, the platform buys data from sellers in exchange of a payment and provides them with differential privacy guarantees. The differential privacy guarantees are determined by optimizing an objective consisting of the misclassification error and the payments. More specifically, our problem has the following components

- We have a set of $m$ sellers, with seller $i$ having data $z^i=(\boldsymbol{x}^i,y^i)$. with $y^i\in\{+1,-1\}$. Therefore, let $D=\{(\boldsymbol{x}^1,y^1),\ldots,(\boldsymbol{x}^m,y^m)\}$ denote the dataset.

- We model the cost that the sellers incur due to loss of privacy using privacy sensitivity $c_i\geq 0$. In other words, if the seller is provided a differential privacy guarantee of $\epsilon_i$, then the seller incurs a total cost of $c_i\cdot u(\epsilon_i)$ where $u(.)$ is considered to be a convex and strictly increasing function with $u(0)=0$. This is consistent with the practical observation that the privacy cost increase of seller $i$ for a slight increase in $\epsilon_i$, will be higher for larger values of $\epsilon_i$. Additionally, the knowledge of the function $u(.)$ is public information.

- Sellers can potentially lie about their privacy sensitivity to get an advantage. Therefore, we denote the reported privacy sensitivity of seller $i$ by $c_i'$. The mechanism will be designed so that $c_i'=c_i$.

- As is standard in the mechanism design literature, we assume that seller $i$'s cost $c_i$ is drawn from a probability density function $f_i(\cdot)$, which is common knowledge. Moreover, we assume that sellers cannot lie about their data. This assumption is valid in scenarios such as healthcare data, where patient information is already within the possession of the hospital. In this context, sellers merely need to grant permission to the hospital (a trusted authority) to utilize their data, specifying their privacy sensitivities in the process.

- The buyer announces a mechanism, i.e., in return for $\{(\boldsymbol{x}^i, y^i), c_i\}$, each seller is guaranteed a differential privacy level $\epsilon_i$ and payment $t_i$ both of which depend on dataset $D$ and reported privacy sensitivities $\boldsymbol{c}'$. [2]

- Based on the privacy loss and the payment received, the cost function of seller $i$ with privacy sensitivity $c_i$, reported privacy sensitivity $c'_i$, and data point $z^i = (\boldsymbol{x}^i, y^i)$ is given by

$$\mathrm{COST}(c_i, \boldsymbol{c}_{-i}, c'_i, \boldsymbol{c}_{-i}; \epsilon_i, t_i) = c_i \cdot u(\epsilon_i) - t_i. \tag{4}$$

To design the mechanism, we next state the objectives of the buyer.

- The buyer learns an ML model $\theta(D, \boldsymbol{c}')$ from dataset $D$, and computes a payment $t_i(D, \boldsymbol{c}')$ to seller $i$ while guaranteeing a privacy level $\epsilon_i(D, \boldsymbol{c}')$ to each seller $i$. To do this, the buyer optimizes a combination of the test loss incurred by the ML model $\mathbb{L}(D, \boldsymbol{c}'; \boldsymbol{\epsilon}, \boldsymbol{\theta})$ and the payments $t_i(D, \boldsymbol{c}')$. The overall objective of the buyer is to minimize

$$\mathbb{E}_{\boldsymbol{c}}\left[\mathbb{L}(D, \boldsymbol{c}'; \boldsymbol{\epsilon}, \boldsymbol{\theta}) + \gamma \sum_i t_i(D, \boldsymbol{c}')\right], \tag{5}$$

where $\gamma$ is a hyperparameter that adjusts the platform's priority to get a better predictor or reduce payments.

- The buyer is also interested in ensuring each seller is incentivized to report their privacy sensitivities truthfully. To that end, the IC property imposes that no seller can misrepresent their privacy sensitivity if others report truthfully, i.e.,

$$\mathrm{COST}(c_i, \boldsymbol{c}_{-i}, c_i, \boldsymbol{c}_{-i}; \epsilon_i, t_i) \leq \mathrm{COST}(c_i, \boldsymbol{c}_{-i}, c'_i, \boldsymbol{c}_{-i}; \epsilon_i, t_i) \quad \forall i, c'_i, \boldsymbol{c}. \tag{6}$$

- Moreover, the buyer wants to ensure that sellers are incentivized to participate. Thus, the IR property imposes the constraint that the platform does not make sellers worse off by participating in the mechanism.

$$\mathrm{COST}(c_i, \boldsymbol{c}_{-i}, c'_i, \boldsymbol{c}_{-i}; \epsilon_i, t_i) \leq 0 \quad \forall i, c'_i, \boldsymbol{c} \tag{7}$$

Using ideas from Myerson (1981) we show that, if $\epsilon_i(D, \boldsymbol{c}')$ is privacy guarantee provided to seller $i$, then using the IC and IR constraints, we can replace the payments $t_i(D, \boldsymbol{c}')$ in the objective function by $\Psi_i(c_i) u\big(\epsilon_i(D, \boldsymbol{c}')\big)$ where $\Psi_i(c) = c + F_i(c)/f_i(c)$ [3] Further, the IC constraint incentivizes sellers to be truthful, and henceforth, we can replace $\boldsymbol{c}'$ with $\boldsymbol{c}$. Since this replacement is a generalization of the payment identity in Fallah et al. (2023) we state and prove this result along with mentioning the required regularity assumptions on $\Psi_i$ in a later section. Substituting it in Eq. (5), we get

$$\min_{\boldsymbol{w}, \boldsymbol{\epsilon}(\cdot)} \mathbb{E}_{\boldsymbol{c}}\left[\mathbb{L}(D, \boldsymbol{c}; \boldsymbol{\epsilon}, \boldsymbol{\theta}) + \gamma \cdot \sum_{i=1}^{m} \Psi_i(c_i) u(\epsilon_i(D, \boldsymbol{c}))\right]. \tag{8}$$

Therefore, buyer's problem reduces to solving Eq. (8) while ensuring $\epsilon_i(D, \boldsymbol{c})$ differential privacy. We refer to Appendix C for a simple numerical example. The order of operations of our mechanism can be summarized as follows.

- The sellers provide the platform with their data $(\boldsymbol{x}_i, y_i)$ and their privacy sensitivity $c_i$.

- The platform announces that in exchange for the data, it will pay according to the payment identity.

- The platform uses this data to obtain an ML model $\theta(D, \boldsymbol{c}')$ and sets privacy levels $\boldsymbol{\epsilon}(\boldsymbol{D}, \boldsymbol{c}')$, i.e., even if the model $\theta(D, \boldsymbol{c}')$ is released publicly, each seller $i$ will be guaranteed differential privacy of $\epsilon_i$.

Note that the payment mechanism does not depend on the choice of loss function. Since we consider our ML problem to be logistic regression the loss function $\mathbb{L}(D, \boldsymbol{c}; \boldsymbol{\epsilon}, \boldsymbol{\theta})$ in our case becomes the misclassification loss $\mathbb{E}[\mathbb{I}_{\{sign(\boldsymbol{w}^T \boldsymbol{x}) \neq y\}}]$, where $\boldsymbol{w}(\boldsymbol{D}, \boldsymbol{c})$ is the weight vector corresponding to the regression model.

---

[2]$\boldsymbol{c} = [c_1, c_2, \ldots, c_m]$. Same notation is used in writing $\boldsymbol{\epsilon}, \boldsymbol{c}', \boldsymbol{t}$.
[3]$F(c_i)$ and $f(c_i)$ denote the values of CDF and PDF functions for privacy sensitivities at $c_i$, respectively.

## 3.1 Calculating the Payments

Before we proceed to solve Eq. (8), we first state the result which calculates the payments based on IC and IR

**Assumption 3.1.** The virtual cost $\Psi_i(c) = c + \frac{F_i(c)}{f_i(c)}$ is an increasing function of $c$.

**Theorem 3.2.** *Assume that $c_i$ is drawn from a known PDF $f(\cdot)$. Given a mechanism design problem with privacy sensitivities $\boldsymbol{c}$ and privacy guarantees $\boldsymbol{\epsilon}$, let the sellers' costs be given by Eq. (4). Then, using the IC and IR constraints, the payments $t_i(\boldsymbol{c})$ can be substituted by $\Psi_i(c_i)u(\epsilon_i(\boldsymbol{c}))$ where $\Psi_i(\boldsymbol{c_i})$ is the virtual cost function given by*

$$\Psi_i(c_i) = c_i + \frac{F_i(c_i)}{f_i(c_i)} \quad \forall i \in \mathbb{N}, c_i \in \mathbb{R} \tag{9}$$

The theorem is a generalization of the payment identity in Fallah et al. (2023). The result is similar to Myerson's interpretation of mechanism design to virtual welfare maximization.

## 3.2 Solving the Mechanism Design Problem

Now, we use the results in previous subsections to solve the mechanism design problem, i.e., Eq. (8). From Theorem 3.2, we obtain the payments, and Eq. (3) provides us with a proxy for the logistic loss, i.e., $\mathbb{E}[\mathbb{I}_{sign(\boldsymbol{w}^T\boldsymbol{x}) \neq y}]$ while also satisfying differential privacy constraints. Thus, our mechanism design objective can be written as

$$\min_{\boldsymbol{a}, \eta, \boldsymbol{w}, \boldsymbol{\epsilon}} \left[ \sum_{i=1}^{m} a_i \log(1 + e^{-y^i \cdot \boldsymbol{w}^T \boldsymbol{x}^i}) + \frac{2\boldsymbol{b}^T \boldsymbol{w}}{\eta} + \frac{\lambda}{2} ||\boldsymbol{w}||^2 + \mu \|\boldsymbol{a}\| + \sigma \frac{1}{\eta} + \gamma \sum_{i=1}^{m} u(\epsilon_i) \Psi_i(c_i) \right],$$

s.t. $\sum_i a_i = 1, \boldsymbol{a} \geq 0, \boldsymbol{a} \leq k/m, \eta \geq 0, a_i \eta \leq \epsilon_i$.

Note that the above objective is minimized with respect to $\boldsymbol{\epsilon}$ when $a_i \eta = \epsilon_i \ \forall i$. Using this equality and $\sum_i a_i = 1$, we get $a_i = \epsilon_i/\eta$, where $\eta = \sum_i \epsilon_i$. Thus, the final objective function for mechanism design can be written as

$$\min_{\boldsymbol{a}, \eta, \boldsymbol{w}} \left[ \sum_{i=1}^{m} a_i \log(1 + e^{-y^i \cdot \boldsymbol{w}^T \boldsymbol{x}^i}) + \frac{2\boldsymbol{b}^T \boldsymbol{w}}{\eta} + \frac{\lambda}{2} ||\boldsymbol{w}||^2 + \mu \|\boldsymbol{a}\| + \sigma \frac{1}{\eta} + \gamma \sum_{i=1}^{m} u(a_i, \eta) \Psi_i(c_i) \right], \tag{10}$$

subject to $\eta > 0, \boldsymbol{a} \leq k/m, \boldsymbol{a} \geq 0$, and $\sum_i a_i = 1$. Here, $\{\beta, \mu, \sigma\}$ are hyperparameters while $\gamma$ is used to tradeoff between test loss and payments. After solving the optimization problem (10), $\boldsymbol{\epsilon}$ can be obtained using $\epsilon_i = a_i \eta$.

**Remark:** The constraint $a_i \leq k/m$ for some $k > 0$ indirectly imposes the condition that $\epsilon_i$ is upper-bounded by a finite quantity. Since $\eta = \sum \epsilon_i$, we can write $\eta = m\epsilon_{avg}$, which together with $\epsilon_i = a_i \eta$ implies $\epsilon_i \leq k\epsilon_{avg}$. Later, we will show that it is sufficient to constraint $\epsilon_{avg}$ to a bounded set while optimizing the objective. Therefore, the upper bound constraint on $a_i$ means that $\epsilon_i$ is upper-bounded. In other words, we can incorporate constraints such as sellers unwilling to tolerate more than a certain amount of privacy loss even if they are paid generously for it.

## 3.3 Interpretation of the terms in the objective

We can divide the objective function Eq. (10) into three parts.

1. The first part is given by $\sum_{i=1}^{m} a_i \log(1 + e^{-y^i \cdot \boldsymbol{w}^T \boldsymbol{x}^i}) + 2\boldsymbol{b}^T \boldsymbol{w}/\eta$. This focuses on obtaining $\boldsymbol{w}$ to solve the differentially-private logistic regression problem.

2. The second part $\mu \|\boldsymbol{a}\| + \sigma/\eta$ denotes the difference between true loss $\mathbb{L}(D, \boldsymbol{c}; \boldsymbol{\epsilon}, \boldsymbol{w})$ and $\hat{\mathbb{L}}(D, \boldsymbol{a}, \boldsymbol{w}, \eta)$. This tries to reduce the gap between $\mathbb{L}(D, \boldsymbol{c}; \boldsymbol{\epsilon}, \boldsymbol{w})$ and $\hat{\mathbb{L}}(D, \boldsymbol{a}, \boldsymbol{w}, \eta)$. We see that the gap reduces as $a_i$ approach $1/m$ and $\eta \to \infty$. Therefore, higher weight on these terms would mean that the optimal solution is forced to pick similar values for $a_i$ and smaller noise, which leads to the standard logistic regression problem.

3. Finally, $\gamma \sum_{i=1}^m u(a_i\eta)\Psi_i(c_i)$ accounts for payments made to sellers. Here, increasing $\gamma$ would mean that the platform would focus more on reducing the payments rather than designing a better logistic regression model.

To gain more insight into the optimal solution of Eq. (10), we first state the first-order necessary conditions. For this analysis, we consider $u(.)$ to be linear, i.e., $u(a_i\eta) = a_i\eta$.

**Theorem 3.3.** *Let* $(\boldsymbol{a^*}, \boldsymbol{w^*}, \eta^*)$ *be the optimal solution for the optimization problem* (10). *Then*

$$a_i^* = \frac{\|\boldsymbol{a^*}\|}{\mu}\left(\tau - \gamma\eta^*\psi_i(c_i) - \log(1 + e^{-y^i\cdot(\boldsymbol{w^*})^T\boldsymbol{x}^i})\right)^+$$

*where* $(f(x))^+ = \max(0, f(x))$, *with* $\tau$ *such that*

$$\sum_{i=1}^m \left(\tau - \gamma\eta^*\psi_i(c_i) - \log(1 + e^{-y^i\cdot(\boldsymbol{w^*})^T\boldsymbol{x}^i})\right)^+ = \frac{\mu}{\|\boldsymbol{a^*}\|}, \tag{11}$$

*Thus,*

$$a_i = \frac{\left(\tau - \gamma\eta^*\psi_i(c_i) - \log(1 + e^{-y^i\cdot(\boldsymbol{w^*})^T\boldsymbol{x}^i})\right)^+}{\sum_{j=1}^m \left(\tau - \gamma\eta^*\psi_j(c_j) - \log(1 + e^{-y^j\cdot(\boldsymbol{w^*})^T\boldsymbol{x}^j})\right)^+}$$

*where* $\eta^*$ *is given by*

$$\eta^* = \left(\frac{\sigma + 2\boldsymbol{b}^T\boldsymbol{w}}{\gamma\sum_{i=1}^m \psi_i(c_i)a_i^*}\right)^{1/2}.$$

From Theorem 3.3, we can make certain observations:

1) First, we note that $a_i$ depends inversely on the privacy sensitivities $c_i$, i.e., $\psi_i(c_i)$ is an increasing function of $c_i$. Therefore, the platform will be willing to buy relatively more privacy from sellers whose per-unit privacy costs are lower. Additionally, the platform will choose not to use data points with excessively high virtual costs.

2) Note that from Eq. (11), $\tau$ is directly proportional to $\mu$ (the weight on $\|\boldsymbol{a}\|$). Further, a higher $\tau$ will reduce the variance in $a_i^*$ because $\tau$ will dominate over $\gamma\eta^*\psi_i(c_i) - \log(1 + e^{-y^i\cdot(\boldsymbol{w^*})^T\boldsymbol{x}^i})$. Therefore, if $\mu \to \infty$ then $\tau \to \infty$ which would make $a_i \to 1/m$. Additionally, by considering a higher value of $\mu$, we can indirectly satisfy the constraint $a_i \le k/m$.

3) Finally, $\eta$ is inversely proportional to $\gamma$. Therefore, a lower weight on payments, i.e., smaller $\gamma$ and thus more focus on getting a better model would mean that the optimal solution will try to reduce noise by making $\eta^* \to \infty$.

### 3.4 Discussion

We make the following remarks on our solution:

- The payment mechanism is independent of $\mathbb{L}(D, \boldsymbol{c}; \boldsymbol{\epsilon}, \boldsymbol{w})$. Thus, using an upper bound for the misclassification loss does not affect the mechanism. Therefore, if one uses a tighter bound, it will help the platform get a better estimator for the same payments. However, it would not affect the behavior of sellers, i.e., sellers will still be incentivized to be truthful and willing to participate in the mechanism.

- Since the payment mechanism does not depend on the choice of the function $\mathbb{L}(D, \boldsymbol{c}; \boldsymbol{\epsilon}, \boldsymbol{w})$, designing payment mechanism and solving objective function can be treated as two separate problems. Thus, any such mechanism design problem can be decoupled into separate problems.

- Finally, we can see that our algorithm can be used to solve the logistic regression problem with heterogeneous privacy guarantee requirements. Previous work in the literature, such as Chaudhuri et al. (2011), solves the problem in the case when it is assumed that all users have the same differential privacy requirements. In our paper, Eq. (3) extends it to the case when users are allowed to have different privacy requirements.

### 3.5 Robustness to correlations between the data points and privacy sensitivities

In our model, we consider that the data point $(\boldsymbol{x}^i, y^i)$ is independent of privacy sensitivity $c_i$. However, in some applications, this might not hold. For example, if $\boldsymbol{x}^i$ is the income of an individual then people with a high income might be reluctant to share their data. Therefore their privacy sensitivities $c_i$ would be higher. This could potentially deter the platform from incorporating data points from high-income individuals, as it would imply higher costs.

Our model, however, remains versatile in handling such practical intricacies, thanks to the presence of the regularization term $\mu\|\boldsymbol{a}\|$. Since, $\|\boldsymbol{a}\|$ is minimized when $a_i = 1/m$, a higher value of $\mu$ will force $a_i$ to be closer to $1/m$. This can also be inferred by the observations made from Theorem 3.3. Therefore, we can always tweak $\mu$ to ensure that all data points are sufficiently considered in the objective therefore helping the platform get a higher classification accuracy while also making sure that the payments are small. The robustness of the model to correlations between $(\boldsymbol{x}^i, y_i)$ and $c_i$ further highlights the importance of adding additional regularization terms which we derived in Thm 2.1.

### 3.6 Asymptotic Analysis

It is also instructive to analyze the objective function in the regime where the number of sellers is large, i.e., when $m \to \infty$. For this purpose, we make the following assumptions:

1. The data set is linearly separable, that is, there exists a $\boldsymbol{w^*}$ such that $\boldsymbol{w^*}^T \boldsymbol{x^i} y^i \geq \delta \ \forall i$ for some $\delta > 0$.

2. $c$ has a bounded support implying that $\psi_i(c_i)$ is bounded. Therefore, let $\psi_i(c_i) \in [p, q]$.

3. There is sufficient finite probability mass around $c = p$ such that the following condition holds: $\exists \ k > 0$, such that $\lim_{m \to \infty} m \cdot \mathbb{P}\big(\psi_i(c_i) \leq p + 1/m^k\big) \to \infty$.

**Theorem 3.4.** *Assume that dataset and privacy sensitivities satisfy (a)-(c) above. Furthermore, let $\|\boldsymbol{b}\| \sim \Gamma(n, 1)$. Then, as $m \to \infty$, the objective function can be upper-bounded almost surely as*

$$\lim_{m \to \infty} \min_{\boldsymbol{w}, \boldsymbol{\epsilon}, \|\boldsymbol{w}\| \leq \beta} \mathbb{E}[\mathbb{I}_{\{sign(\boldsymbol{w}^T \boldsymbol{x}) \neq y\}}] + \gamma \sum_{i=1} \epsilon_i \Psi_i(c_i) \leq \log(1 + e^{-\frac{\delta}{\sqrt{p\gamma}}}) + 2\sqrt{\sigma p\gamma + 2\|\boldsymbol{b}\|\sqrt{p\gamma}} \tag{12}$$

*wherein the payment is at most $\sqrt{\sigma p\gamma + 2\|\boldsymbol{b}\|\sqrt{p\gamma}}/\gamma$. In particular, the inequality is non-trivial if $p\gamma$ satisfies*

$$\log(1 + e^{-\frac{\delta}{\sqrt{p\gamma}}}) + 2\sqrt{\sigma p\gamma + 2\|\boldsymbol{b}\|\sqrt{p\gamma}} < 1.$$

*Furthermore, as $p \to 0$, the above limit becomes zero.*

The first term $\log(1 + e^{-\frac{\delta}{\sqrt{p\gamma}}})$ represents maximum possible error for logistic loss. The second term is extra error due to payment costs and errors associated with ensuring differential privacy. This is unavoidable because there is finite cost ($\psi_i(c_i) \geq p$) associated with each data point.

We observe a dynamic interplay: as $p$ decreases, the cost per data point diminishes, leading to a reduction in payments. Notably, as $p \to 0$ and $m \to \infty$, the upper bound on the error can be driven to 0. This phenomenon is intuitively explained by the platform's ability to select a lot of samples with nearly zero virtual cost from a large pool, enabling the reduction of misclassification error.

# 4 Algorithmic Considerations

This section will discuss algorithmic considerations in solving the optimization problem associated with our mechanism design solution.

## 4.1 Making the objective function convex

The objective functions for logistic regression with heterogeneous differential privacy, i.e., Eq. (3) and for solving the mechanism design problem, i.e., Eq. (10) are nonconvex in $(\boldsymbol{a}, \boldsymbol{w})$. Therefore, we introduce a change of variables trick to make the function convex. We will first prove the convexity result for Eq. (3) and then argue that it also holds for Eq. (10). We make the substitution $a_i = e^{z_i}$. With the proposed modifications, the logistic regression objective in Eq. (3) becomes

$$\min_{\boldsymbol{z}, \eta, \boldsymbol{w}} f(\boldsymbol{w}, \boldsymbol{z}, \eta),$$

$$f(\boldsymbol{w}, \boldsymbol{z}, \eta) = \left[ \sum_{i=1}^{m} e^{z_i} \log(1 + e^{-y^i \cdot \boldsymbol{w}^T \boldsymbol{x}^i}) + \frac{\lambda}{2} \|\boldsymbol{w}\|^2 + \frac{2\boldsymbol{b}^T \boldsymbol{w}}{\eta} + \mu \|e^{\boldsymbol{z}}\| + \sigma \frac{1}{\eta} \right] \tag{13}$$

The following theorem states that the new objective is strongly convex in $(\boldsymbol{w}, \boldsymbol{z})$ for sufficiently large $\lambda$, and thus gradient descent converges exponentially to global infimum.

**Theorem 4.1.** *Given a classification task, let $D$ be a set of data points from $m$ users with $\|\boldsymbol{x}^i\| \leq 1$, for each $i$. Then, there exists a value $\lambda_{conv}$ such that the objective function as defined in Eq. (13) is convex in $(\boldsymbol{w}, \boldsymbol{z})$ for $\lambda > \lambda_{conv}$ and $\mu, \sigma, \eta \in \mathbb{R}_+$. Let $(\boldsymbol{w}^t, \boldsymbol{z}^t)_{t \in \mathbb{N}}$ be the sequence of iterates on applying projected gradient descent on $f(\cdot)$ for a fixed $\eta$ on a convex set $S$, and let $f_\eta^* = \inf_{\boldsymbol{w}, \boldsymbol{z}} f(\boldsymbol{w}, \boldsymbol{z}, \eta)$. Then, for $\lambda > \lambda_{conv}$, there exists $0 < \alpha < 1$, such that*

$$f(\boldsymbol{w}^t, \boldsymbol{z}^t, \eta) - f_\eta^* \leq \alpha^t (f(\boldsymbol{w}^0, \boldsymbol{z}^0, \eta) - f_\eta^*).$$

We also observe experimentally that the projected gradient descent on $(\boldsymbol{w}, \boldsymbol{z})$ for a fixed $\eta$ converges to the same stationary point for different initializations for the real dataset considered in this paper. This suggests that the condition on $\lambda$ in Theorem 4.1 is not very restrictive.

**Remark:** Note that the same change of variables and adding a regularizer for $\boldsymbol{w}$ also makes the mechanism design objective Eq. (10) convex in $(\boldsymbol{w}, \boldsymbol{z})$ for $\lambda > \lambda_{conv}$. This is because $\gamma \sum_{i=1}^{m} u(\eta e^{z_i}) \psi_i(c_i)$ is convex for a fixed $\eta$.

## 4.2 Algorithm

Let us denote the mechanism design objective (10) with the change of variables by

$$g(\boldsymbol{w}, \boldsymbol{z}, \eta) = f(\boldsymbol{w}, \boldsymbol{z}, \eta) + \gamma \sum_{i=1}^{m} u(\eta e^{z_i}) \Psi_i(c_i).$$

To optimize this objective, we first optimize $g(\boldsymbol{w}, \boldsymbol{z}, \eta)$ with respect to $(\boldsymbol{w}, \boldsymbol{z})$ for a fixed $\eta$ over the constraint set using projected gradient descent, and then perform a line search over the scalar parameter $\eta$. Now, to determine the range of $\eta$, note that $\eta = \sum_{i=1}^{m} \epsilon_i = m\epsilon_{avg}$, where $\epsilon_{avg} = \sum_i \epsilon_i / m$. Considering that, in practice, the differential privacy guarantees $\boldsymbol{\epsilon}$ cannot be excessively high, we restrict the range of $\boldsymbol{\epsilon}$ by taking $\epsilon_{avg} \in [0, L]$ for some $L \in \mathbb{R}_+$. Thus, we can choose different values of $\eta$ by discretizing $[0, L]$ to any required precision and choosing $\epsilon_{avg}$ from it. The pseudocode for the algorithm is provided in the provided table.

As a result, for each combination of $\{\lambda, \mu, \sigma, \gamma\}$, the algorithm provides the corresponding optimal weight vector $\boldsymbol{w}$ and privacy guarantees $\boldsymbol{\epsilon}$. The privacy guarantees are then used to determine payments. Moreover, the misclassification error is computed over a validation dataset using $\boldsymbol{w}$. Therefore, we fix $\gamma$ and optimize our objective $\mathbb{E}[\mathbb{I}_{\{sign(\boldsymbol{w}^T \boldsymbol{x}) \neq y\}}] + \gamma \sum_i t_i(D, \boldsymbol{c}')$ wrt $\{\lambda, \mu, \sigma\}$. Subsequently, the platform can pick appropriate value of $\gamma$ by comparing different combinations of payment sum $(\sum_i t_i(D, \boldsymbol{c}'))$ and misclassification loss $(\mathbb{E}[\mathbb{I}_{\{sign(\boldsymbol{w}^T \boldsymbol{x}) \neq y\}}])$ corresponding to each value of $\gamma$.

**Remark:** Note that the same algorithm can be used to solve the logistic regression problem, i.e., Eq. (3) by considering the projection set $S$ to be $\{\boldsymbol{z} : \sum e^{z_i} = 1, \eta e^{z_i} \le \epsilon_i\}$.

---

### An Iterative Algorithm to Optimize the Mechanism Design Objective

Set step-size $\alpha \in (0, 1]$, $g_{\min} = \infty$.
Discretize $[0, L]$ to any required precision and choose $\epsilon_{\text{avg}}$ from this discrete set.
Sample $\|\boldsymbol{b}\|$ from $\Gamma(n, 1)$ with its direction chosen uniformly at random.
**foreach** $\epsilon_{avg}$ **do**
    Initialize $\boldsymbol{w}$ from $\mathbb{N}(0, 1)$, $z_i = \log(\frac{1}{m})$ $\forall i$;
    **while** *not converged* **do**
        $\boldsymbol{w} \leftarrow \boldsymbol{w} - \alpha \frac{d}{d\boldsymbol{w}} g(\boldsymbol{w}, \boldsymbol{z}, \epsilon_{\text{avg}})$,
        $\boldsymbol{z} \leftarrow \boldsymbol{z} - \alpha \frac{d}{d\boldsymbol{z}} g(\boldsymbol{w}, \boldsymbol{z}, \epsilon_{\text{avg}})$,
        $\boldsymbol{z} \leftarrow \text{Proj}_S(\boldsymbol{z})$ s.t. $S = \{\boldsymbol{z} : \sum_{i=1}^{m} e^{z_i} = 1\}$.
    **end**
    **if** $g_{\min} > g(\boldsymbol{w}, \boldsymbol{z}, \epsilon_{avg})$ **then**
        $\boldsymbol{w}_{\text{opt}} \leftarrow \boldsymbol{w}$,
        $\boldsymbol{z}_{\text{opt}} \leftarrow \boldsymbol{z}$,
        $\epsilon_{\text{opt}} \leftarrow \epsilon_{\text{avg}}$,
        $g_{\min} \leftarrow g(\boldsymbol{w}, \boldsymbol{z}, \epsilon_{\text{avg}})$
    **end**
**end**

---

### 4.3 Assumption on Validation data

In addition to $(\boldsymbol{w}, \boldsymbol{z})$, our objective function needs to be minimized on a set of parameters $\{\lambda, \mu, \sigma, \gamma\}$. Therefore, we require a validation dataset to compare different values of the hyperparameters. Since hyperparameters are chosen based on the validation set, if the validation dataset is private, it may violate differential privacy guarantees. Chaudhuri et al. (2011) provides a detailed discussion on ensuring differential privacy using validation data. Referring to their work, we assume existence of a small publicly available dataset that can be used for validation. This assumption ensures that differential privacy guarantees are not violated.

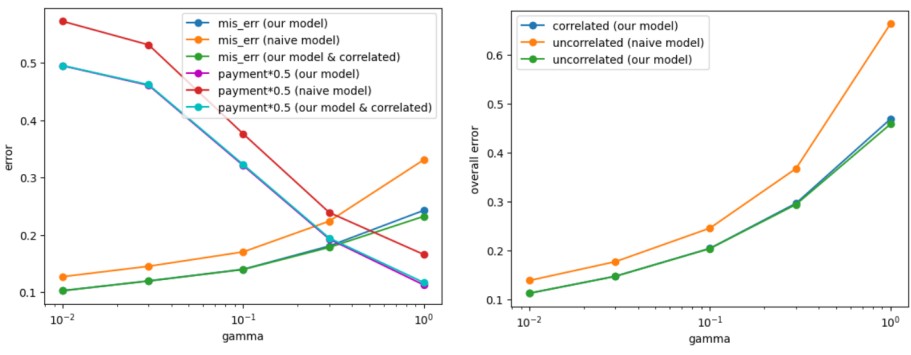

Figure 2: a) Misclassification error and payments b) Comparison of overall error

## 5 Numerical Results

### 5.1 Application to Medical Data

**Dataset and Specifications:** To demonstrate the applications of our proposed mechanism, we perform our mechanism design approach on the Wisconsin Breast Cancer dataset UCI (1995). Furthermore, $c$ is drawn from $\mathbb{U}[e^{-4}, 5e^{-4}]$ and $\psi(c)$ is calculated accordingly to be $2c - e^{-4}$. Also, we consider $u(x) = x$.
**Implementation:** The optimization of the loss function in Eq. (10) is conducted on the training data, and subsequently, the hyperparameters $\{\lambda, \mu, \sigma\}$ are selected based on the validation data for each value of $\gamma$. Therefore, the corresponding misclassification error (misclassified samples/total samples) and payments are plotted for each $\gamma$ in Fig. 2(a). The values are plotted by taking the mean over 15 different samples of the noise vector $b$. Given that our approach is first to consider the tradeoff between payments and model accuracy for ML models, there is a lack of existing methods in the literature for direct comparison. However, to showcase increase in efficiency of our approach due to addition of extra regularization terms $(\mu||a||, \sigma/\eta)$, we compare our results with a *naive model* whose objective does not consider these terms, i.e. Eq. (10) with $\mu = \sigma = 0$. Finally, all results are benchmarked with the baseline error which is misclassification error of the model in absence of payments and differential privacy guarantees. Additionally, to evaluate the efficiency of all the methods the overall error (misclassification error + $\gamma$*payments) is plotted in Fig. 2(b). It is important to note that additional experiments in the appendix provide further insight about the hyperparameters.
**Observations and Practical Usage:** As depicted in Fig. 2(a), there is a tradeoff between misclassification loss and payments, with an increase in misclassification loss and a decrease in payments as $\gamma$ rises. Consequently, the platform can tailor $\gamma$ based on its requirements. For example, if the platform has a budget constraint, the platform can iteratively adjust $\gamma$ to obtain the optimal estimator within the given budget. Finally, from Fig. 2(b), we see that the incorporation of regularization terms $(\mu||a||, \sigma/\eta)$ in the model yields a more efficient mechanism with a lower overall error.
**Robustness to Correlations:** We repeat the above experiment by adding correlations between the data points and their corresponding privacy sensitivities. This is done by mapping elements in $c$ and datapoints using a predefined rule. Specifically, $c$ is sampled and its elements are sorted, while datapoints are sorted based on one of their indices. Consequently, $k^{th}$ datapoint is mapped with $k^{th}$ element of $c$. These observations are plotted in Fig. 2 and which shows that performance of our algorithm is similar, affirming robustness of our approach to correlations between datapoints and privacy sensitivities. This underscores the adaptability and efficacy of our method even in scenarios where correlations are introduced, further validating its practical utility.

## 6 Conclusion

We introduce a novel algorithm to design a mechanism that balances competing objectives: achieving a high-quality logistic regression model consistent with differential privacy guarantees while minimizing payments made to data providers. Notably, our result in Thm. 2.1 can extend to scenarios where individual data points require different weights in loss calculations. Such weighting enables accommodation of noisy measurements or varying costs associated with sample retrieval. Additionally, we note that our model considers heterogeneous privacy guarantees, acknowledging the diverse privacy needs of individuals. Finally, through Thm. 3.2 we see that the payment mechanism does not depend on choice of loss function. Therefore designing a payment mechanism and minimizing the objective can be effectively decoupled and treated as separate problems. This observation along with Thm. 2.1 which highlights the necessity of additional regularization terms opens avenues for design of mechanisms tailored to ML problems of higher complexity.

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

## A    Appendix A: Omitted Proofs

In this section of the appendix, we provide the omitted proofs. We start with the following proposition, whose proof is inspired by that in Chaudhuri et al. (2011).

*Proposition* 1. For any $(\boldsymbol{a}, \eta) \in \mathbb{F}$, the output of Algorithm 1, denoted by $\hat{w}$, preserves $\epsilon$ differential privacy.

**Proof:** Let $\boldsymbol{d}$ and $\boldsymbol{d}'$ be two vectors over $\mathbb{R}^n$ with norm at most 1, and $y$, $y'$ being either $-1$ or 1. Consider two different inputs given by $D = \{(\boldsymbol{x}^1, y^1), \ldots, (\boldsymbol{x}^{m-1}, y^{m-1}), (\boldsymbol{d}, y)\}$ and $D' = \{(\boldsymbol{x}^1, y^1), \ldots, (\boldsymbol{x}^{m-1}, y^{m-1}), (\boldsymbol{d}', y')\}$. Since the function $\hat{\mathbb{L}}(D, \boldsymbol{w}, \boldsymbol{a}, \eta)$ is strictly convex in $\boldsymbol{w}$, for every $\boldsymbol{b}' = \frac{2\boldsymbol{b}}{\eta}$, there is a unique output $\hat{\boldsymbol{w}}$ for each input. Let us denote the values of $\boldsymbol{b}'$ for the first and the second input such that the optimal solution is $\hat{\boldsymbol{w}}$ by $\boldsymbol{b}'_1$ and $\boldsymbol{b}'_2$, respectively, with the corresponding densities $h(\boldsymbol{b}'_1)$ and $h(\boldsymbol{b}'_2)$. We know that the derivative is 0 at $\hat{\boldsymbol{w}}$. Thus, we have

$$\boldsymbol{b}'_1 - \frac{a_m \cdot \boldsymbol{d} y}{1 + e^{y \hat{\boldsymbol{w}}^T \boldsymbol{d}}} = \boldsymbol{b}'_2 - \frac{a_m \cdot \boldsymbol{d}' y'}{1 + e^{y' \hat{\boldsymbol{w}}^T \boldsymbol{d}'}}.$$

Since $\frac{1}{1+e^{y \hat{\boldsymbol{w}}^T \boldsymbol{d}}} < 1$ and $\frac{1}{1+e^{y' \hat{\boldsymbol{w}}^T \boldsymbol{d}'}} < 1$, we have $\|\boldsymbol{b}'_1 - \boldsymbol{b}'_2\| < 2a_m$, which implies $\big|\|\boldsymbol{b}'_1\| - \|\boldsymbol{b}'_2\|\big| < 2a_m$. Therefore, for any pairs $(\boldsymbol{d}, y)$, $(\boldsymbol{d}', y')$, and any set $V \subset \mathbb{R}^n$, we can write

$$\frac{\mathbb{P}[\boldsymbol{w}(\boldsymbol{x}^1, \ldots, \boldsymbol{x}^m = \boldsymbol{d}, y^1, \ldots, y^m = y) \in V]}{\mathbb{P}[\boldsymbol{w}(\boldsymbol{x}^1, \ldots, \boldsymbol{x}^m = \boldsymbol{d}', y^1, \ldots, y^m = y') \in V]} = \frac{h(\boldsymbol{b}'_1)}{h(\boldsymbol{b}'_2)} \cdot \frac{|det(J(\hat{\boldsymbol{w}} \to \boldsymbol{b}'_1|D))|^{-1}}{|det(J(\hat{\boldsymbol{w}} \to \boldsymbol{b}'_2|D'))|^{-1}}$$

where $J(\hat{\boldsymbol{w}} \to \boldsymbol{b}'_1|D)$ is the Jacobian matrix of the mapping from the space of $\boldsymbol{w}$ to $\boldsymbol{b}$.

We first bound the ratio of the determinants. By taking the gradient at $\hat{\boldsymbol{w}}$ to be 0, we have

$$\boldsymbol{b}'_1 = \sum_i \frac{a_i \cdot \boldsymbol{x}^i y^i}{1 + e^{y^i \hat{\boldsymbol{w}}^T \boldsymbol{x}^i}} - \lambda \hat{\boldsymbol{w}}$$

Thus, taking the gradient of $\boldsymbol{b}'_1$ with respect to $\hat{\boldsymbol{w}}$, we have

$$\frac{\delta \boldsymbol{b}'_1}{\delta \hat{\boldsymbol{w}}} = \sum_i \frac{-a_i e^{y^i \hat{\boldsymbol{w}}^T \boldsymbol{x}^i} \cdot \boldsymbol{x}^i (\boldsymbol{x}^i)^T}{(1 + e^{y^i \hat{\boldsymbol{w}}^T \boldsymbol{x}^i})^2} - \lambda I \tag{14}$$

We define two matrices $A$ and $E$ such that

$$A = \sum_i \frac{a_i e^{y^i \hat{\boldsymbol{w}}^T \boldsymbol{x}^i} \cdot \boldsymbol{x}^i (\boldsymbol{x}^i)^T}{(1 + e^{y^i \hat{\boldsymbol{w}}^T \boldsymbol{x}^i})^2} + \lambda I \tag{15}$$

$$E = \frac{-a_m e^{y^m \hat{\boldsymbol{w}}^T \boldsymbol{d}'^m} \cdot \boldsymbol{d}' (\boldsymbol{d}')^T}{(1 + e^{y^m \hat{\boldsymbol{w}}^T \boldsymbol{d}'})^2} - \frac{-a_m e^{y^m \hat{\boldsymbol{w}}^T \boldsymbol{d}^m} \cdot \boldsymbol{d} (\boldsymbol{d})^T}{(1 + e^{y^m \hat{\boldsymbol{w}}^T \boldsymbol{d}})^2} \tag{16}$$

Now,

$$\frac{|det(J(\hat{\boldsymbol{w}} \to \boldsymbol{b}'_1|D))|^{-1}}{|det(J(\hat{\boldsymbol{w}} \to \boldsymbol{b}'_2|D'))|^{-1}} = \frac{|det(A + E)|}{|det(A)|} \tag{17}$$

Let $\lambda_1(M)$ and $\lambda_2(M)$ denote the first and second largest eigenvalues of a matrix $M$. Since, $E$ is of rank 2, from Lemma 10 of Chaudhuri et al. (2011), we have

$$\frac{|det(A + E)|}{|det(A)|} = |1 + \lambda_1(A^{-1}E) + \lambda_2(A^{-1}E) + \lambda_1(A^{-1}E)\lambda_2(A^{-1}E)| \tag{18}$$

Now, since we consider logistic loss which is convex, any eigenvalue of A is atleast $\lambda$. Thus $|\lambda_j(A^{-1}E)| \le \frac{1}{\lambda}|\lambda_j(E)|$. Now, applying the triangle inequality to the trace norm, we have

$$|\lambda_1(E)| + |\lambda_2(E)| \le 2\|d\|^2 a_m \frac{e^{y^m \hat{\boldsymbol{w}}^T \boldsymbol{d}^m}}{(1 + e^{y^m \hat{\boldsymbol{w}}^T \boldsymbol{d}^m})^2} \le 2a_m \tag{19}$$

Therefore by AM-GM inequality $\lambda_1(E)\lambda_2(E) \le a_m^2$

Thus

$$\frac{|det(A + E)|}{|det(A)|} \le \left(1 + \frac{a_m}{\lambda}\right)^2 \tag{20}$$

Therefore, we have

$$
\begin{aligned}
\frac{\mathbb{P}[\boldsymbol{w}(\boldsymbol{x}^1, \ldots, \boldsymbol{x}^m = \boldsymbol{d}, y^1, \ldots, y^m = y) \in V]}{\mathbb{P}[\boldsymbol{w}(\boldsymbol{x}^1, \ldots, \boldsymbol{x}^m = \boldsymbol{d}', y^1, \ldots, y^m = y') \in V]} &\leq e^{\eta(\|\boldsymbol{b'}_1\| - \|\boldsymbol{b'}_2\|)/2} \cdot \frac{|det(A+E)|}{|det(A)|} \leq e^{a_m \eta} \leq e^{\epsilon_m}, \\
&\leq \exp(a_m \eta + 2\log(1 + \frac{a_m}{\lambda})) \\
&\leq \exp\left(a_m \eta + 2\log\left(1 + \frac{k}{m\lambda}\right)\right) \\
&\leq \exp\left(\epsilon_m + 2\log\left(1 + \frac{k}{m\lambda}\right)\right) \leq \exp(\epsilon_m + \Delta) \quad (21)
\end{aligned}
$$

where the last inequality holds because $(\boldsymbol{a}, \eta) \in \mathbb{F}$. Note that, we also consider that $a_m$ are constrained such that $a_m \leq \frac{k}{m}$ for some $k > 0$. Note that

∎

**Theorem A.1.** *Given a classification task, let $D$ be the dataset from $m$ users with $\|\boldsymbol{x}^i\| \leq 1 \; \forall i$. Further, consider that users have differential privacy requirements $\boldsymbol{\epsilon} = (\epsilon_i)_{i=1}^m \in \mathbb{R}_+^m$, respectively. Also, let $\mathbb{L}(D, \boldsymbol{c}; \boldsymbol{\epsilon}, \boldsymbol{w}) = \mathbb{E}[\mathbb{I}_{\{sign(\boldsymbol{w}^T \boldsymbol{x}) \neq y\}}]$ be misclassification loss and $\hat{\mathbb{L}}(D, \boldsymbol{w}, \boldsymbol{a}, \eta)$ be as defined in Algorithm 1. Then, the following holds with probability at least $(1 - \delta)(1 - \delta')$ for every $\boldsymbol{\epsilon} \in \mathbb{R}^m$ and $(\boldsymbol{a}, \eta) \in \mathbb{F}$:*

$$
\sup_{\|\boldsymbol{w}\| \leq \beta} \left| \mathbb{L}(D, \boldsymbol{c}; \boldsymbol{\epsilon}, \boldsymbol{w}) - \hat{\mathbb{L}}(D, \boldsymbol{a}, \boldsymbol{w}, \eta) \right| \leq \mu \|a\| + \frac{\sigma}{\eta}.
$$

*Thus, the misclassification loss can be upper-bounded by*

$$
\mathbb{L}(D, \boldsymbol{c}; \boldsymbol{\epsilon}, \boldsymbol{w}) \leq \hat{\mathbb{L}}(D, \boldsymbol{a}, \boldsymbol{w}, \eta) + \mu \|a\| + \frac{\sigma}{\eta}. \quad (22)
$$

$\forall \boldsymbol{w} \; s.t. \; \|\boldsymbol{w}\| \leq \beta, \; (\boldsymbol{a}, \eta) \in \mathbb{F}$.

**Proof:** For any sample $S = \{(\boldsymbol{x}^1, y^1), \ldots, (\boldsymbol{x}^m, y^m)\}$ and any $\boldsymbol{w} \in \mathbb{R}^n$, we define the empirical loss function as

$$
\hat{\mathbb{L}}_S[\boldsymbol{w}] = \sum_{i=1}^m a_i \log(1 + e^{-y^i \cdot \boldsymbol{w}^T \boldsymbol{x}^i}) + \boldsymbol{b'}^T \boldsymbol{w},
$$

where $\|\boldsymbol{b'}\| \sim \Gamma(n, \frac{2}{\eta})$, and the direction of $\boldsymbol{b'}$ is chosen uniformly at random. The true loss function is given by

$$
\mathbb{E}[\mathbb{I}_{\{sign(\boldsymbol{w}^T \boldsymbol{x}) \neq y\}}] \leq \mathbb{E}[\log(1 + e^{-y \cdot \boldsymbol{w}^T \boldsymbol{x}})] = \mathbb{L}[\boldsymbol{w}].
$$

Since $\sum_i a_i = 1$ and $\mathbb{E}[\boldsymbol{w}] = 0$, we have

$$
\mathbb{E}\left[\hat{\mathbb{L}}_S[\boldsymbol{w}]\right] = \sum_i a_i \mathbb{E}[\log(1 + e^{-y \cdot \boldsymbol{w}^T \boldsymbol{x}})] + \mathbb{E}[\boldsymbol{b'}^T \boldsymbol{w}]
$$

$$
= \mathbb{E}[\log(1 + e^{-y \cdot \boldsymbol{w}^T \boldsymbol{x}})] = \mathbb{L}[\boldsymbol{w}]. \quad (23)
$$

Let $\phi(S) = \sup_{\boldsymbol{w} \in \mathbb{R}^n}(\mathbb{L}[\boldsymbol{w}] - \hat{\mathbb{L}}_S[\boldsymbol{w}])$. To bound $\left|\boldsymbol{b'}^T \boldsymbol{w}\right|$, we consider the event where $\left|\boldsymbol{b'}^T \boldsymbol{w}\right| < r$. This should be true for all $\boldsymbol{w}$ such that $\|\boldsymbol{w}\| \leq \beta$. This is possible only when $\|\boldsymbol{b'}\| < r/\beta$. Consider that this event happens with probability $1 - \delta'$. From the CDF of $\Gamma(n, \frac{2}{\eta})$, we get

$$
\sum_{i=0}^{n-1} \frac{(\frac{\eta r}{2\beta})^i}{i!} e^{-\frac{\eta r}{2\beta}} = \delta'. \quad (24)
$$

Let $\eta r / \beta = t$ and $v(t) = \sum_{i=0}^{n-1} \frac{(t/2)^i}{i!} e^{-\frac{t}{2}}$. Also, it is known that $v(t)$ is a monotonously decreasing function. Therefore, its inverse exists, and we have

$$
r = \frac{\beta v^{-1}(\delta')}{\eta}.
$$

Now, using McDiarmid's inequality,

$$P(|\phi(S) - \mathbb{E}_S[\phi(S)]| > t) \leq \exp\left(\frac{-2t^2}{\sum a_i^2 \log^2(1 + e^\beta) + (\frac{2\beta v^{-1}(\delta')}{\eta})^2}\right).$$

Thus, with probability at least $(1 - \delta)(1 - \delta')$,

$$\phi(S) \leq \mathbb{E}_S[\phi(S)] + \sqrt{\frac{\ln\frac{1}{\delta}\left(\sum a_i^2 \log^2(1 + e^\beta) + (\frac{2\beta v^{-1}(\delta')}{\eta})^2\right)}{2}}. \tag{25}$$

Moreover, we can write

$$\mathbb{E}_S[\phi(S)] = \mathbb{E}_S\left[\sup_{\|w\| \leq \beta}(\mathbb{L}[\boldsymbol{w}] - \hat{\mathbb{L}}_S(\boldsymbol{w}))\right]$$

$$= \mathbb{E}_S\left[\sup_{\|w\| \leq \beta}\mathbb{E}_{S'}[\hat{\mathbb{L}}_{S'}(\boldsymbol{w}) - \hat{\mathbb{L}}_S(\boldsymbol{w})]\right]$$

$$= \mathbb{E}_{S,S'}\left[\sup_{\|w\| \leq \beta}[\hat{\mathbb{L}}_{S'}(\boldsymbol{w}) - \hat{\mathbb{L}}_S(\boldsymbol{w})]\right]$$

$$= \mathbb{E}_{S,S',\sigma_i}\left[\sup_{\|w\| \leq \beta}\left[\sum_{i=1}^m a_i\sigma_i\left(\log(1 + e^{-y'^i \cdot \boldsymbol{w}^T \boldsymbol{x}'^i}) - \log(1 + e^{-y^i \cdot \boldsymbol{w}^T \boldsymbol{x}^i})\right)\right]\right]$$

$$\leq \mathbb{E}_{S,\sigma_i}\left[\sup_{\|w\| \leq \beta}\left[\sum_{i=1}^m a_i\sigma_i\left(\log(1 + e^{-y^i \cdot \boldsymbol{w}^T \boldsymbol{x}^i})\right)\right]\right] + \mathbb{E}_{S',\sigma_i}\left[\sup_{\|w\| \leq \beta}\left[\sum_{i=1}^m -a_i\sigma_i\left(\log(1 + e^{-y'^i \cdot \boldsymbol{w}^T \boldsymbol{x}'^i})\right)\right]\right]$$

$$= 2R_m(\boldsymbol{w}),$$

where in the above derivations, $R_m(\boldsymbol{w})$ is given by

$$R_m(\boldsymbol{w}) = \mathbb{E}_{\sigma,S}\left[\sup_{\|w\| \leq \beta}\sum_{i=1}^m a_i\sigma_i \log(1 + e^{-y^i \cdot \boldsymbol{w}^T \boldsymbol{x}^i})\right],$$

and the fourth equality is obtained by introducing uniformly independent random variables $\sigma_i$ taking values in $\{-1, 1\}$. Now, we can again use McDiarmid's inequality to get

$$R_m(\boldsymbol{w}) \leq \hat{R}_S(\boldsymbol{w}) + \sqrt{\frac{\ln\frac{1}{\delta}\left(\sum_i a_i^2 \log^2(1 + e^\beta) + (\frac{2\beta v^{-1}(\delta')}{\eta})^2\right)}{2}}$$

$$\leq \hat{R}_S(\boldsymbol{w}) + \sqrt{\frac{\ln\frac{1}{\delta}}{2}}\left(\sqrt{\sum a_i^2 \log^2(1 + e^\beta)} + \frac{2\beta v^{-1}(\delta')}{\eta}\right).$$

Finally, we calculate $\hat{R}_S(\boldsymbol{w})$ as

$$\hat{R}_S(\boldsymbol{w}) = \mathbb{E}_\sigma\left[\sup_{\boldsymbol{w} \in R^n}\sum_i a_i\sigma_i \log(1 + e^{-y^i \boldsymbol{w}^T \boldsymbol{x}^i}) + \boldsymbol{b'}^T \boldsymbol{w}\right]$$

$$\leq \frac{1}{\ln 2}E_\sigma\left[\sup_{\boldsymbol{w}}\sum_i a_i\sigma_i(-y^i \boldsymbol{w}^T \boldsymbol{x}^i)\right] + \frac{\beta v^{-1}(\delta')}{\eta}$$

$$\leq \frac{\|\boldsymbol{w}\|}{\ln 2}\mathbb{E}_\sigma\left[\sum_i a_i\sigma_i \boldsymbol{x}^i\right] + \frac{\beta v^{-1}(\delta')}{\eta}$$

$$\leq \frac{\beta}{\ln 2}\sqrt{\sum_i a_i^2} + \frac{\beta v^{-1}(\delta')}{\eta},$$

where the first inequality holds by the Lipschitz property, and the last inequality uses $\|\boldsymbol{x}\| \le 1$ and $\|\boldsymbol{w}\| \le \beta$. Putting it together, we have

$$
\mathbb{E}[\mathbb{I}_{\{sign(\boldsymbol{w}^T\boldsymbol{x}) \neq y\}}] \le \sum_{i=1}^{m} a_i \log(1 + e^{-y^i \boldsymbol{w}^T \boldsymbol{x}^i})) + \boldsymbol{b'}^T \boldsymbol{w} + \left[\left(\frac{3 \ln \frac{1}{\delta}}{\sqrt{2}}\right) \log(1 + e^\beta) + \frac{\beta}{\ln 2}\right] \sqrt{\sum_i a_i^2}
$$
$$
+ \left(\frac{6 \ln \frac{1}{\delta}}{\sqrt{2}} + 1\right)\left(\frac{2\beta v^{-1}(\delta')}{\eta}\right) + \frac{\lambda}{2}\|\boldsymbol{w}\|^2
$$

Thus, it is enough to define $\mu(\delta, \beta) = \left(\frac{3 \ln \frac{1}{\delta}}{\sqrt{2}}\right) \log(1 + e^\beta) + \frac{\beta}{\ln 2}$ and $\sigma(\delta, \delta', \beta) = \left(\frac{6 \ln \frac{1}{\delta}}{\sqrt{2}} + 1\right)\left(2\beta v^{-1}(\delta')\right)$. ∎

**Assumption A.2.** The virtual cost $\Psi_i(c) = c + \frac{F_i(c)}{f_i(c)}$ is an increasing function of $c$.

**Theorem A.3.** *Assume that $c_i$ is drawn from a known PDF $f(\cdot)$. Given a mechanism design problem with privacy sensitivities $\boldsymbol{c}$ and privacy guarantees $\boldsymbol{\epsilon}$, let the sellers' costs be given by Eq. (4). Then, using the IC and IR constraints, the payments $t_i(\boldsymbol{c})$ can be substituted in the objective by $\Psi_i(c_i)u(\epsilon_i(\boldsymbol{c}))$ where $\Psi_i(\boldsymbol{c_i})$ is the virtual cost function given by*

$$
\Psi_i(c_i) = c_i + \frac{F_i(c_i)}{f_i(c_i)} \quad \forall i \in \mathbb{N}, c_i \in \mathbb{R} \tag{26}
$$

**Proof:** The proof follows similar steps as those in Fallah et al. (2023). Let $h_i(c) = \mathbb{E}_{\boldsymbol{c}_{-i}}[\mathbb{L}(D, \boldsymbol{c}; \boldsymbol{\epsilon}, \boldsymbol{\theta})]$, where $c$ is the argument corresponding to the privacy of agent $i$. Similarly, let $t_i(c) = \mathbb{B}_{\boldsymbol{c}_{-i}}[t_i(D, c, \boldsymbol{c}_{-i})]$ and $u(\epsilon_i(c)) = \mathbb{E}_{\boldsymbol{c}_{-i}}[u(\epsilon_i(D, c, \boldsymbol{c}_{-i}))]$. Using the IC constraint, we have

$$
c_i \cdot u(\epsilon_i(c_i)) - t_i(c_i) \le c_i \cdot u(\epsilon_i(c_i')) - t_i(c_i').
$$

From the IC constraint, the function $c_i \cdot u(\epsilon_i(c)) - t_i(c)$ has a minima at $c = c_i$. Thus, by equating the derivative to 0 and substituting $c = c_i$, we get

$$
c_i \cdot \left(\frac{du(\epsilon_i(c))}{dc}\right)_{c=c_i} = t_i'(c_i) \tag{27}
$$

Solving for $c_i$ from this equation we get,

$$
t_i(c_i) = t_i(0) + c_i u(\epsilon_i(c_i)) - \int_0^{c_i} u(\epsilon_i(z))dz. \tag{28}
$$

If an individual does not participate in estimating the parameter by not giving their data, then their loss function will be 0. Thus, using the IR constraint, for all $c_i$ we have

$$
t_i(0) \ge \int_0^{c_i} u(\epsilon_i(z))dz.
$$

Because $u(\epsilon_i(c_i)) \ge 0$, it implies

$$
t_i(0) \ge \int_0^\infty u(\epsilon_i(z))dz
$$

Plugging this relation into Eq (28), we get

$$
t_i(c_i) \ge c_i u(\epsilon_i(c_i)) + \int_{c_i} u(\epsilon_i(z))dz.
$$

Thus, for given $\boldsymbol{c}$, the payments are calculated to be $c_i u(\epsilon_i(c_i)) + \int_{c_i} u(\epsilon_i(z))dz$. Also, note that the minimum cost required for $c_i = \infty$ would be 0. Therefore, this can also be written as $-\int_{c_i} z \frac{d}{dz} u(\epsilon_i(z))dz$. This is an

interesting observation because the payment obtained in our problem is similar to the Myerson's payment mechanism. Now, we can compute $\mathbb{E}_{c_i}[t_i(c_i)]$ as

$$\mathbb{E}_{c_i}[t_i(c_i)] = \mathbb{E}_{c_i}[c_i u(\epsilon_i(c_i))] + \mathbb{E}_{c_i}\Big[\int_{c_i} u(\epsilon_i(z))dz\Big]$$

$$= \int_{z_{-i}} \int_{z_i} \Big(z_i u(\epsilon_i(z_i, z_{-i})) + \int_{y_i=z_i} u(\epsilon_i(y, z_{-i}))dy_i\Big) f_i(z_i)dz_i f_{-i}(z_{-i})dz_{-i}.$$

By changing the order of integrals, we have

$$\mathbb{E}_{c_i}[t_i(c_i)] = \mathbb{E}_{\boldsymbol{c}}[\Psi_i(c_i)u(\epsilon_i(\boldsymbol{c}))],$$

where $\Psi_i(c_i) = c_i + \frac{F_i(c_i)}{f_i(c_i)}$. Therefore, to minimize the expected error, for any given $\boldsymbol{c'}$, one can choose

$$t_i(D, \boldsymbol{c'}) = \Psi_i(c_i)u(\epsilon_i(D, \boldsymbol{c'})),$$

which completes the proof. ∎

**Theorem A.4.** *Let $(\boldsymbol{a^*}, \boldsymbol{w^*}, \eta^*)$ be the optimal solution for the objective function given in Eq. (10). Then* [4]

$$a_i^* = \Big(\tau - \gamma\eta^*\psi_i(c_i) - \log(1 + e^{-y^i \cdot (\boldsymbol{w^*})^T \boldsymbol{x}^i})\Big)^+ \Big(\frac{\|\boldsymbol{a^*}\|}{\mu}\Big),$$

*with $\tau$ such that*

$$\sum_{i=1}^m \Big(\tau - \gamma\eta^*\psi_i(c_i) - \log(1 + e^{-y^i \cdot (\boldsymbol{w^*})^T \boldsymbol{x}^i})\Big)^+ = \frac{\mu}{\|\boldsymbol{a^*}\|},$$

*where $\eta^*$ is given by*

$$\eta^* = \Big(\frac{\sigma + 2\boldsymbol{b}^T \boldsymbol{w}}{\gamma \sum_{i=1}^m \psi_i(c_i)a_i^*}\Big)^{1/2}.$$

**Proof:** The Lagrangian for the objective function (10) is given by

$$\min_{\boldsymbol{a}, \eta, \boldsymbol{w}} \sum_{i=1}^m a_i \log(1 + e^{-y^i \cdot \boldsymbol{w}^T \boldsymbol{x}^i}) + \frac{\lambda}{2}\|\boldsymbol{w}\|^2 + \frac{2\boldsymbol{b}^T \boldsymbol{w}}{\eta} + \mu\|\boldsymbol{a}\| + \sigma\frac{1}{\eta} + \gamma\eta \sum a_i \Psi_i(c_i)$$

$$+ \tau(1 - \sum_{i=1}^m a_i) - \sum_{i=1}^m \zeta_i a_i - \kappa\eta.$$

Now if $(\boldsymbol{a^*}, \boldsymbol{w^*}, \eta^*)$ is the optimal solution, then it should satisfy the first order necessary conditions. Therefore, by taking the derivatives of the Lagrangian function with respect to $\boldsymbol{a}$ and $\eta$ and setting them to 0, we get

$$a_i^* = \Big(\tau - \gamma\eta^*\psi_i(c_i) - \log(1 + e^{-y^i \cdot (\boldsymbol{w^*})^T \boldsymbol{x}^i})\Big)^+ \Big(\frac{\|\boldsymbol{a^*}\|}{\mu}\Big),$$

$$\eta^* = \Big(\frac{\sigma + 2\boldsymbol{b}^T \boldsymbol{w}}{\gamma \sum_{i=1}^m \psi_i(c_i)a_i^*}\Big)^{1/2}.$$

Finally, using the constraint $\sum_{i=1}^m a_i = 1$, we obtain

$$\sum_{i=1}^m \Big(\tau - \gamma\eta^*\psi_i(c_i) - \log(1 + e^{-y^i \cdot (\boldsymbol{w^*})^T \boldsymbol{x}^i})\Big)^+ = \frac{\mu}{\|\boldsymbol{a^*}\|}.$$

∎

---

[4] $(f(x))^+$ is used to denote $\max(0, f(x))$.

**Theorem A.5.** *Assume that dataset and privacy sensitivities satisfy conditions (a)-(c) given in section 3.6. Furthermore, let $\|\boldsymbol{b}\| \sim \Gamma(n, 1)$. Then, as $m \to \infty$, there exists a constant $d > 0$ such that the objective function can be upper-bounded almost surely as*

$$\lim_{m\to\infty} \min_{\boldsymbol{w},\boldsymbol{\epsilon},\|\boldsymbol{w}\|\leq\beta} \mathbb{E}[\mathbb{I}_{\{sign(\boldsymbol{w}^T\boldsymbol{x})\neq y\}}] + \gamma \sum_{i=1} \epsilon_i \Psi_i(c_i)$$

$$\leq \log(1 + e^{-\frac{\delta}{\sqrt{p\gamma}}}) + 2\sqrt{\sigma p\gamma + 2\|\boldsymbol{b}\|\sqrt{p\gamma}},$$

*wherein the payment is at most $\sqrt{\sigma p\gamma + 2\|\boldsymbol{b}\|\sqrt{p\gamma}}/\gamma$. In particular, the inequality is non-trivial if $p\gamma$ satisfies*

$$\log(1 + e^{-\frac{\delta}{\sqrt{p\gamma}}}) + 2\sqrt{\sigma p\gamma + 2\|\boldsymbol{b}\|\sqrt{p\gamma}} < 1.$$

*Furthermore, as $p \to 0$, the above limit becomes zero.*

**Proof:** We first consider the case where $p > 0$. We can write

$$\lim_{m\to\infty} \min_{\boldsymbol{a},\eta,\boldsymbol{w},\|w\|\leq\beta,\beta} \left[ \sum_{i=1}^m a_i \log(1 + e^{-y^i\cdot\boldsymbol{w}^T\boldsymbol{x}^i}) + \frac{2\boldsymbol{b}^T\boldsymbol{w}}{\eta} + \mu\|\boldsymbol{a}\| + \sigma\frac{1}{\eta} \right] + \gamma\eta \sum_{i=1}^m a_i \Psi_i(c_i)$$

$$\leq \lim_{m\to\infty} \min_{\beta,\eta} \left[ \log(1 + e^{-\delta\|x\|\cdot\|w\|}) + \|\boldsymbol{b}\|\frac{2\beta}{\eta} + \mu\|\boldsymbol{a}\| + \sigma\frac{1}{\eta} \right] + \gamma\eta \sum a_i \Psi_i(c_i)$$

$$\leq \lim_{m\to\infty} \min_{\beta,\eta} \left[ \log(1 + e^{-\delta\beta}) + \|\boldsymbol{b}\|\frac{2\beta}{\eta} + \mu\|\boldsymbol{a}\| + \sigma\frac{1}{\eta} \right] + \gamma\eta \sum a_i \Psi_i(c_i)$$

$$\leq \min_{\beta} \log(1 + e^{-\delta\beta}) + \sqrt{\sigma + 2\beta\|\boldsymbol{b}\|}\sqrt{p\gamma}$$

$$\leq \log(1 + e^{-\frac{\delta}{\sqrt{p\gamma}}}) + \sqrt{\sigma p\gamma + 2\|\boldsymbol{b}\|\sqrt{p\gamma}} \tag{29}$$

Let us choose $a_i = \frac{1}{N}$ for $\psi_i(c_i) \leq p + \frac{1}{m^k}$ for some $k > 0$, where $N$ is a random variable denoting the number of datapoints for which $\psi_i(c_i) \leq p + \frac{1}{m^k}$. Therefore, $N \to m\mathbb{P}(\psi_i(c_i) \leq p + \frac{1}{m^k})$ almost surely. Thus, $\lim_{m\to\infty} \|a\| = 0$. Further, take $\eta = \frac{\sqrt{(\sigma+2\beta\|b\|)}}{\sqrt{p\gamma}}$. For the last step, we set $\beta = \frac{1}{\sqrt{p\gamma}}$. The minimum value of the function will be smaller than this particular choice of variables. Note that a trivial solution can be $\eta = 0$. Also, misclassification loss can be at most 1. Therefore,

$$\lim_{p\to 0} \lim_{m\to\infty} \min_{\substack{\boldsymbol{w},\boldsymbol{\epsilon}\\\beta,\|\boldsymbol{w}\|\leq\beta}} \mathbb{E}[\mathbb{I}_{\{sign(\boldsymbol{w}^T\boldsymbol{x})\neq y\}}] + \gamma \sum_{i=1}^m \epsilon_i \Psi_i(c_i) \leq 1.$$

Thus, Eq. (29) is non-trivial when $p\gamma$ is such that

$$\log(1 + e^{-\frac{\delta}{\sqrt{p\gamma}}}) + \sqrt{\sigma p\gamma + 2\|b\|\sqrt{p\gamma}} < 1.$$

For $p \to 0$, the above loss converges to 0. Therefore,

$$\lim_{p\to 0} \lim_{m\to\infty} \min_{\boldsymbol{w},\boldsymbol{\epsilon}} \mathbb{E}[\mathbb{I}_{\{sign(\boldsymbol{w}^T\boldsymbol{x})\neq y\}}] + \gamma \sum_{i=1}^m \epsilon_i \Psi_i(c_i)$$

$$\leq \lim_{p\to 0} \lim_{m\to\infty} \min_{\boldsymbol{a},\eta,\boldsymbol{w},\|w\|\leq\beta,\beta} \left[ \sum_{i=1}^m a_i \log(1 + e^{-y^i\cdot\boldsymbol{w}^T\boldsymbol{x}^i}) + \frac{2\boldsymbol{b}^T\boldsymbol{w}}{\eta} + \mu\|\boldsymbol{a}\| + \sigma\frac{1}{\eta} \right] + \gamma\eta \sum a_i \Psi_i(c_i) = 0.$$

Next, we consider the case of $p = 0$. We choose $a_i = \frac{1}{N}$ for $\psi_i(c_i) \leq \frac{1}{m^k}$ for some $k > 0$. Therefore, $N \to m\mathbb{P}(\psi_i(c_i) \leq \frac{1}{m^k})$ almost surely. Thus, $\lim_{m\to\infty} \|a\| = 0$. Further, take $\eta = \frac{1}{m^{k'}}$, where $0 < k' < k$, and

$\beta = m^{k''}$, where $0 < k'' < k'$. By substituting these parameters into the above expression, we get

$$\lim_{m \to \infty} \min_{\substack{\boldsymbol{a}, \eta, \boldsymbol{w} \\ \|w\| \le \beta, \beta}} \left[ \sum_{i=1}^{m} a_i \log(1 + e^{-y^i \cdot \boldsymbol{w}^T \boldsymbol{x}^i}) + \frac{2\boldsymbol{b}^T \boldsymbol{w}}{\eta} + \mu \|\boldsymbol{a}\| + \sigma \frac{1}{\eta} + \gamma\eta \sum_{i=1}^{m} a_i \Psi_i(c_i) \right]$$

$$\le \lim_{m \to \infty} \log(1 + e^{-\delta m^{k''}}) + \frac{2||b||m^{k''}}{m^{k'}} + \sigma \frac{1}{m^{k'}} + \gamma \frac{m^{k'}}{m^k}$$

$$= 0.$$

This completes the proof. ∎

**Theorem A.6.** *Given a classification task, let $D$ be a set of data points from $m$ users with $\|\boldsymbol{x}^i\| \le 1$, for each $i$. Then, there exists a value $\lambda_{conv}$ such that the objective function as defined in Eq. (13) is convex in $(\boldsymbol{w}, \boldsymbol{z})$ for $\lambda > \lambda_{conv}$ and $\mu, \sigma, \eta \in \mathbb{R}_+$. Let $(\boldsymbol{w}^t, \boldsymbol{z}^t)_{t \in \mathbb{N}}$ be the sequence of iterates on applying projected gradient descent on $f(\cdot)$ for a fixed $\eta$ on a convex set $S$, and let $f_\eta^* = \inf_{\boldsymbol{w}, \boldsymbol{z}} f(\boldsymbol{w}, \boldsymbol{z}, \eta)$. Then, for $\lambda > \lambda_{conv}$, there exists $0 < \alpha < 1$, such that*

$$f(\boldsymbol{w}^t, \boldsymbol{z}^t, \eta) - f_\eta^* \le \alpha^t (f(\boldsymbol{w}^0, \boldsymbol{z}^0, \eta) - f_\eta^*).$$

**Proof:** We prove that the function

$$\sum_{i=1}^{m} \left[ e^{z_i} \log(1 + e^{-\boldsymbol{w}^T \boldsymbol{x}^i \cdot y^i}) + \frac{\lambda_0^i}{2} \cdot \|\boldsymbol{w}\|^2 + \gamma \cdot m\epsilon_{avg} e^{z_i} \Psi_i \right],$$

where $\sum_{i=1}^{m} \lambda_0^i = \lambda$, is jointly convex in $\boldsymbol{w}$ and $\boldsymbol{z}$, which implies that the objective function will also be jointly convex. The Hessian matrix for the $i^{th}$ term of the loss function takes the form of

$$\begin{bmatrix} a & b(x_1^i y^i) & \dots & b(x_n^i y^i) \\ b(x_1^i y^i) & 2\lambda_0^i + c(x_1^i y^i)^2 & \dots & c(x_1^i y^i)(x_n^i y^i) \\ b(x_2^i y^i) & c(x_1^i y^i)(x_2^i y^i) & \dots & c(x_2^i y^i)(x_n^i y^i) \\ . & . & \dots & . \\ . & . & \dots & . \\ . & . & \dots & . \\ b(x_n^i y^i) & c(x_1^i y^i)(x_n^i y^i) & \dots & c(x_n^i y^i)^2 + 2\lambda_0^i \end{bmatrix},$$

where $a$, $b$, and $c$ in the above matrix are given by

$$a = e^{z_i} \log(1 + e^{-\boldsymbol{w}^T \boldsymbol{x}^i \cdot y^i}),$$

$$b = e^{z_i} \frac{e^{-\boldsymbol{w}^T \boldsymbol{x}^i y^i}}{1 + e^{-\boldsymbol{w}^T \boldsymbol{x}^i y^i}} \frac{1}{\ln 2},$$

$$c = e^{z_i} \frac{e^{-\boldsymbol{w}^T \boldsymbol{x}^i y^i}}{(1 + e^{-\boldsymbol{w}^T \boldsymbol{x}^i y^i})^2} \frac{1}{\ln 2}.$$

Using column elimination, we get

$$\begin{bmatrix} a & b(x_1^i y^i) & 0 & \dots & 0 \\ b(x_1^i y^i) & 2\lambda_0^i + c(x_1^i y^i)^2 & -2\lambda_0^i \frac{x_2^i y^i}{x_1^i y^i} & \dots & -2\lambda_0^i \frac{x_n^i y^i}{x_1^i y^i} \\ b(x_2^i y^i) & c(x_1^i y^i)(x_2^i y^i) & 2\lambda_0^i & \dots & 0 \\ . & . & . & \dots & . \\ . & . & . & \dots & . \\ . & . & . & \dots & . \\ b(x_n^i y^i) & c(x_1^i y^i)(x_n^i y^i) & 0 & \dots & 2\lambda_0^i \end{bmatrix}.$$

Thus, the determinant is equal to

$$
a \cdot \begin{bmatrix}
2\lambda_0^i + c(x_1^i y^i)^2 & -2\lambda_0^i \frac{x_2^i y^i}{x_1^i y^i} & -2\lambda_0^i \frac{x_3^i y^i}{x_1^i y^i} & \dots & -2\lambda_0^i \frac{x_n^i y^i}{x_1^i y^i} \\
c(x_1^i y^i)(x_2^i y^i) & 2\lambda_0^i & 0 & \dots & 0 \\
\cdot & \cdot & \cdot & \dots & \cdot \\
\cdot & \cdot & \cdot & \dots & \cdot \\
\cdot & \cdot & \cdot & \dots & \cdot \\
c(x_1^i y^i)(x_n^i y^i) & 0 & 0 & \dots & 2 * \lambda_0^i
\end{bmatrix}
$$

$$
-b(x_1^i y^i) \begin{bmatrix}
b(x_1^i y^i) & -2\lambda_0^i \frac{x_2^i y^i}{x_1^i y^i} & -2\lambda_0^i \frac{x_3^i y^i}{x_1^i y^i} & \dots & -2\lambda_0^i \frac{x_n^i y^i}{x_1^i y^i} \\
b(x_2^i y^i) & 2\lambda_0^i & 0 & \dots & 0 \\
\cdot & \cdot & \cdot & \dots & \cdot \\
\cdot & \cdot & \cdot & \dots & \cdot \\
\cdot & \cdot & \cdot & \dots & \cdot \\
b(x_n^i y^i) & 0 & 0 & \dots & 2\lambda_0^i
\end{bmatrix}.
$$

The first matrix (whose determinant is multiplied by $a$ ) is of the form $A + 2\lambda_0^i I$, where $A$ is a rank 1 matrix. Thus, $A$ has $(n-1)$ eigenvalues of 0, and we calculate the non-zero eigenvalue to be $c\|x\|^2$. Hence, its determinant equals $(2\lambda_0^i)^{(n-1)}(2\lambda_0^i + c\|x\|^2)$. Expanding the second term, we get $(2\lambda_0^i)^{(n-1)}b^2\|x\|^2$. Thus, to ensure that the Hessian is positive definite, we need

$$
2\lambda_0^i > \frac{b^2 - ac}{a}\|x^i\|^2
$$

$$
\Rightarrow 2\lambda_0^i > \|x^i\|^2 \left(\frac{1}{\ln 2}\right)^2 \cdot e^{z_i} \left(\frac{e^{-w^T x^i y^i}}{1 + e^{-w^T x^i y^i}}\right)^2 \left(\frac{1}{\log(1 + e^{-w^T x^i y^i})} - \ln 2 \cdot e^{w^T x^i y^i}\right).
$$

Thus, the coefficient of $\|w\|^2$ (denoted by $\lambda$) needs to be

$$
2\lambda > \sum_i e^{z_i}\|x^i\|^2 \left(\frac{1}{\ln 2}\right)^2 \cdot \left(\frac{e^{-w^T x^i y^i}}{1 + e^{-w^T x^i y^i}}\right)^2 \left(\frac{1}{\log(1 + e^{-w^T x^i y^i})} - \ln 2 \cdot e^{w^T x^i y^i}\right),
$$

$$
\Rightarrow 2\lambda > \left(\frac{1}{\ln 2}\right)^2 \cdot \max_i \|x^i\|^2 \left(\frac{e^{-w^T x^i y^i}}{1 + e^{-w^T x^i y^i}}\right)^2 \left(\frac{1}{\log(1 + e^{-w^T x^i y^i})} - \ln 2 \cdot e^{w^T x^i y^i}\right). \tag{30}
$$

Next, we proceed to prove the second part. First, observe that for $\lambda > \lambda_{conv}$, the function is $(\lambda - \lambda_{conv})$-strictly convex. Let the function be $\mu$-strictly convex. Denote $(w, z)$ by $v$, and let $\inf_v f(v) = L$. Then, there exists $v^* \in \mathbb{R}^{m+n}$, such that $\forall \delta > 0$

$$
f(v^*) - L < \delta
$$

Thus, we have

$$
f(v) - L = f(v) - f(v^*) + f(v^*) - L
$$
$$
\leq \delta + \langle \nabla f(v), v - v^* \rangle - \frac{\mu}{2}\|v^* - v\|^2
$$
$$
\leq \delta + \frac{1}{2\mu}\|\nabla f(v)\|^2. \tag{31}
$$

Next, we prove that $f$ is $L$-smooth. Let $\nabla_w f(v)$ and $\nabla_z f(v)$ denote the gradient vectors of $f$ with respect to $w$ and $z$, respectively. Then

$$
\|\nabla f(v^1) - \nabla f(v^2)\|^2
$$
$$
= \|\nabla_w f(v^1) - \nabla_w f(v^2)\|^2 + \|\nabla_z f(v^1) - \nabla_z f(v^2)\|^2
$$
$$
= \|\nabla_w f(v_1) - \nabla_w f(v_2)\|^2 + \|e^{z^1}(\log(1 + e^{-w^{1T} xy}) + \frac{e^{z^1}}{\|(e^{z^1})\|} - e^{z^2}(\log(1 + e^{-w^{2T} xy}) + \frac{e^{z^2}}{\|(e^{z^2})\|}\|^2.
$$

Now, the logistic loss is $L_1$-smooth for some $L_1 > 0$. Moreover, since $\|x\|, \|w\|$ is bounded, $\|\log(1 + e^{-w^T xy})\|$ is bounded. Further, $\|(1/\|(e^z)\|)\| < \sqrt{m}$. Thus, there exists an $K$ such that $f$ is $K$-smooth.

Consider that the step size for gradient descent is chosen such that, $\gamma K \leq 1$. Therefore, by descent lemma, we have

$$f(v^{t+1}) \leq f(v^t) + \langle \triangledown f(v^t), v^{t+1} - v^t \rangle + \frac{K}{2}\|v^{t+1} - v^t\|^2$$

$$\leq f(v^t) - \gamma\|\triangledown f(v^t)\|^2 + \frac{K\gamma^2}{2}\|\triangledown f(v^t)\|^2$$

$$= f(v^t) - \frac{\gamma}{2}(2 - K\gamma)\|\triangledown f(v^t)\|^2$$

$$\leq f(v^t) - \frac{\gamma}{2}\|\triangledown f(v^t)\|^2.$$

The second inequality uses the condition for projection, and that projection is non-expansive. Finally, the update rule is substituted to obtain terms with $\|\triangledown f(v^t)\|^2$. By using Eq (31) and applying recursion, we get

$$f(v^t) - L \leq (1 - \gamma\mu)^t(f(v^0) - L) + \delta.$$

∎

## B  Appendix B: Additional Experiments

In this appendix, we perform additional experiments on generated synthetic data to demonstrate more trends in the solution. For synthetic data, we consider the classification boundary to be a linear separator passing through the origin. Input data $\boldsymbol{x}^i$ is generated by sampling from i.i.d. zero-mean Gaussian distribution with bounded variance. Corresponding outputs, $y^i$, are generated using the linear separator. Furthermore, $\boldsymbol{c}$ is drawn from $\mathbb{U}[p, q]$, where $p, q \in \mathbb{R}$. Moreover, unless stated otherwise, we consider the same hyperparameter values as in the case of real data, and $\gamma$ is taken as 1.0.

**Intuition behind hyperparameter** $\mu$

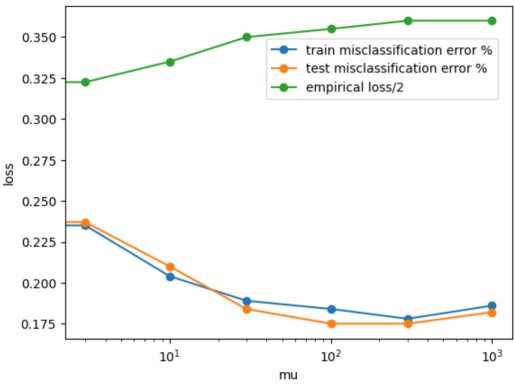

Figure 3: Algorithm performance with respect to $\eta$.

**Experiment Details:** We start with validating our earlier point about the importance of considering generalization loss in the objective function. For this, we solve the logistic regression problem while ensuring heterogeneous differential privacy, i.e., Eq. (3). The misclassification error is compared for different values of $\mu$. A higher value of $\mu$ means a higher focus on generalization error. Averaged results for different values of the differential privacy guarantee $\boldsymbol{\epsilon}$ and noise vector $\boldsymbol{b}$ are plotted in Figure 3. Train/test misclassification error is the percentage of misclassified samples while empirical loss is given by $\sum_{i=1}^m a_i \log(1 + e^{-y^i \cdot \boldsymbol{w}^T \boldsymbol{x}^i})$.
**Observations:** We see that as $\mu$ increases, there is a reduction in both train and test misclassification errors at first, and then it increases slightly. Therefore, choosing the correct value of $\mu$ is important to achieve the best classification accuracy. Moreover, we observe that empirical loss increases as $\mu$ increases. This means

that for a smaller $\mu$, the empirical loss is small even if samples are misclassified. Therefore, optimizing over the empirical loss alone might not result in a good logistic regression model. Hence, it is also necessary to consider generalization terms in the optimization.

**Performance with respect to distribution of $c$**

In order to observe the effect of the distribution of $c$ on the solution, we vary $p$ and $q$ where $c \sim \mathbb{U}[p, q]$. The results are illustrated in Figure 4. As expected, it is observed that the variance in $a$ (weight attached to each datapoint) decreases as the variance of $c$ decreases. Moreover, as the variance of $c$ goes to 0, the variance of $a$ also vanishes. This implies that, as the cost per unit loss of privacy (i.e., privacy sensitivity of all the users) becomes the same, the optimal choice for the platform is to treat each data point almost equally by providing them the same privacy guarantee while performing logistic regression.

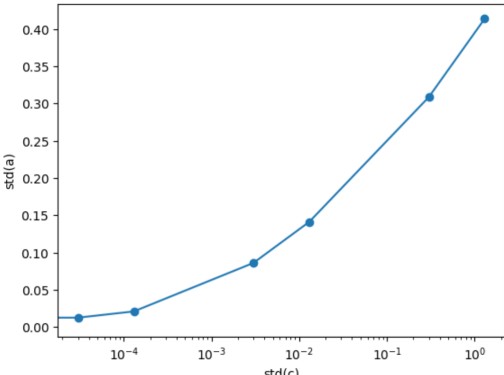

Figure 4: Effect of the variance of $c$ on the variance of $a$.

**Performance with respect to parameter $\eta$**

Here, we show how the addition of noise affects the performance of logistic regression. We fix $a_i = \frac{1}{m}$ and vary $\eta$. We write $\eta = m \cdot \epsilon_{avg}$ and vary $\epsilon_{avg}$. Therefore, we solve logistic regression by minimizing the following objective while adding noise to ensure differential privacy.

$$\min_{\boldsymbol{w}} \left[ \sum_{i=1}^{m} a_i \log(1 + e^{-y^i \cdot \boldsymbol{w}^T \boldsymbol{x}^i}) + \frac{\lambda}{2} \|\boldsymbol{w}\|^2 + \frac{2\boldsymbol{b}^T \boldsymbol{w}}{\eta} \right],$$

As before, $\|\boldsymbol{b}\| \sim \Gamma(n, 1)$ and its direction is chosen uniformly at random. Note that this has been done in Chaudhuri et al. (2011). However, we performed the experiments and show our observations to show completeness. The experiments are repeated for different values of $\boldsymbol{b}$, and the observations are averaged and depicted in Figure 5.

It can be seen that when $\eta$ increases (and, therefore, the amount of noise decreases), the algorithmic performance increases, as expected. This is seen through a decrease in both train and test misclassification errors.

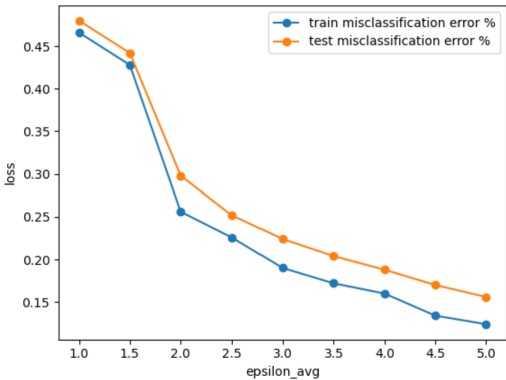

Figure 5: Algorithm performance with respect to $\eta$.

## C  Appendix C: A Simple Example

Consider a simple problem of solving logistic regression using a dataset collected from privacy sensitive sellers. Also consider that $u(x) = x$. Therefore, the buyer's objective is as before

$$\mathbb{E}_{\boldsymbol{c}}\left[\mathbb{E}[\mathbb{I}_{\{sign(\boldsymbol{w}^T\boldsymbol{x})\neq y\}}] + \gamma \sum_i t_i\right] \tag{32}$$

Thus, the problem is formulated as follows

- Let the input be one-dimensional and the dataset consist of two sellers. Therefore let $D = \{(\boldsymbol{x}^1, y^1), (\boldsymbol{x}^2, y^2)\} = \{(1, 1), (-1, -1)\}$.

- Moreover, let the privacy sensitivities be $c_1 = 0.1, c_2 = 0.6$. Further, assume that the privacy sensitivities are iid and come from $\mathbb{U}[0, 1]$.

- Therefore, our goal is to find the optimal model weights $\boldsymbol{w}$, differential privacy guarantees $\epsilon_1, \epsilon_2$ and the payments $t_1, t_2$. Additionally, $\boldsymbol{w}$ needs to be calculated such that it is consistent with $\epsilon_1, \epsilon_2$.

To solve this, we will first calculate $\psi_i$. For $\boldsymbol{c} \sim \mathbb{U}[0, 1]$

$$\psi_i = c_i + F(c_i)/f(c_i) = 2c_i \tag{33}$$

Thus, $\psi_1(c_1) = 0.2, \psi_2(c_2) = 1.2$

Therefore, we calculate $\boldsymbol{w}, \epsilon_1, \epsilon_2$ by optimizing the below equation

$$\min_{\boldsymbol{a}, \eta, \boldsymbol{w}} \left[ \sum_{i=1}^{m} a_i \log(1 + e^{-y^i \cdot \boldsymbol{w}^T \boldsymbol{x}^i}) + \frac{2\boldsymbol{b}^T\boldsymbol{w}}{\eta} + \mu\|\boldsymbol{a}\| + \sigma\frac{1}{\eta} + \gamma\eta \sum_{i=1}^{m} a_i \Psi_i(c_i) \right], \tag{34}$$

For different values of $\{\mu, \sigma, \gamma\}$ we get corresponding values of $\boldsymbol{w}, \epsilon_1, \epsilon_2$. Next, $\boldsymbol{w}$ is evaluated on a validation dataset to get the misclassification error $\mathbb{E}_{\boldsymbol{c}}\left[\mathbb{E}[\mathbb{I}_{\{sign(\boldsymbol{w}^T\boldsymbol{x})\neq y\}}]\right]$. Moreover, we use the payment identity to get the payments corresponding to $\epsilon_i$. Finally, the best combination of payments and model accuracy is selected based on the platform's needs.

