# OpenReview forum: "Striking a Balance: An Optimal Mechanism Design for Heterogenous Differentially Private Data Acquisition for Logistic Regression"
_TMLR — Rejected by TMLR_

### Review · Reviewer_rRmq · 2024-05-06

**Summary Of Contributions:**

This papers proposes a new differentially private mechanism, that solves logistic regression under heterogeneous differential privacy. The problem is worded as a "market" problem, where a "platform" (that does the training) buys data records from "sellers" (that own the data; although data is stored by the platform). A loss function is designed to solve logistic regression with heterogeneous privacy budget, and heterogeneous "privacy sensitivity", allowing to collect more data from sellers whose privacy sensitivity is lower (that is, seller who require lower payment for their data). An adapted loss, that account for the impact of handling heterogeneous privacy on generalization, is proposed, as well as a convex variant of that loss which allows to speed-up the computation.

**Audience:**

Yes

**Broader Impact Concerns:**

The point of view that is defended in this paper implies that data owners may be willing to trade some privacy for better payments/utility. This can a be a bit dangerous, in a world where the poor could be tempted to sell out their privacy in exchange for some "payment" (whatever that could be). This is particularly concerning as this goal is explicitely stated as the second contribution in page 2.

The considered setting, where a trusted party holds all the data, and asks for permission from data owners to use the data in exchange of a payment, is not always realistic. In pratice, it may be the case that people hold their own data locally, and are only willing to share data after a privacy-preserving procedure has been used. Most results presented in this paper would not apply in such a setting.

**Claims And Evidence:**

No

**Requested Changes:**

1. The remark at the end of Section 3.2 states that hard constraint on privacy parameters $\epsilon_i$'s can be enforced. However, the proposed bound is uniform over $i \in \\{1, \dots, m\\}$: what happens if one seller has very strong privacy requirements? Does it forces everyone's privacy parameter to be as low as their parameter? A clear result on the actual privacy guarantees for each individual would be a remarkable addition to current results.
2. I find the definitions of $a_i$ and $\eta$ are a bit confusing. I wonder if the exposition would be more clear if replacing $a_i$ by $a_i = \alpha_i \epsilon_i$ with a constraint on the coefficient $\alpha_i \in [0, 1]$. Then, everything could be stated in terms of $\alpha_i$, that could be interpreted as the "amount of the privacy budget used for the computation" for a given client.
3. The term "privacy sensitivity" may not be the best choice, as "sensitivity" is often used in the design of differentially private mechanisms to refer to the sensitivity of a function (which is used to calibrate the noise).
4. Although not of critical interest, are the results difficult to adapt to a setting where data owners have multiple records?

**Strengths And Weaknesses:**

**Strengths**
1. An interesting point view on data collection in differentially privacy machine learning is proposed, making the trade-off between privacy sensitivity and the return one may obtain from sharing its data explicit.
2. The proposed method gives heterogeneous privacy guarantees, allowing to handle different data owners with different privacy sensitivities by using a reweighted loss, that accounts for the impact of this reweighting on generalization error.
3. An efficient algorithm for solving the proposed problem is given.

**Weaknesses**
1. There are no utility guarantees on the proposed method: one could expect to see a utility result as it is classical in the differentially private optimization literature. While this is not necessarily a problem, and could be compensated by numerical evaluation, the method is only evaluated on a single dataset, with a very restricted range of sensitivities $c$.
2. Study of the impact of correlation (Section 3.5) is very limited; one could expect to see the behavior of the proposed method for different levels of correlation between features, labels and privacy sensitivities.
3. The IC and IR properties are stated in Section 3, and are then used for computing the payment. However, it seems they are rather used as *assumptions* rather than actual properties of the proposed mechanisms. More details on why the proposed mechanism enjoys these properties would be nice. In particular, these properties are used to claim that data owners would not lie on their true privacy sensitivity, which in the end sounds more like a hidden assumption than a property of the mechanism.

---

> ### Author Response · Authors · 2024-05-28
> **Response by the Authors**
>
> We thank the reviewer for their review and their comments. Please find our response to the requested changes, weaknesses, and broader impact concerns below
>
> Response to Weaknesses:
>
> 1)  Since the goal of the platform is to minimize the misclassification accuracy of the classifier along with minimizing the payments, we first note that there is no exact analytical representation for the expected misclassification accuracy of the machine learning model. This forces us to use upper bounds on the misclassification accuracy for our analysis. Therefore, since there is no exact representation of the misclassification accuracy, obtaining utility results is difficult. However, in our paper, we provide an analysis of our results in the asymptotic case when the number of samples goes to infinity.
>
> 2) We would like to make sure that we understand their comment. As far as we can tell and as acknowledged by Reviewer 1, we have a fairly general model of the correlation between data and privacy sensitivity. Therefore, we interpret the reviewer's comment as one requesting more simulations with different levels of correlation. We can indeed add more simulations exploring the impact of correlation in the revised version of the paper.
>
> 3) The IC and IR properties mentioned in the paper are imposed as conditions that the proposed payment needs to satisfy. In the proof of theorem 3.2 (page 18), the IC and IR conditions are imposed on the payments and the resulting payment identity obtained thus satisfies these properties.
>
> Response to the Requested Changes:
>
> 1) The current model in our paper considers a fixed bound on the privacy parameters which can then be handled by limiting the range of $\epsilon_{avg}$. The idea behind this is that often the platform needs to also adhere to certain privacy regulations/laws for example, the General Data Protection Regulation (GDPR) in the European Union or the California Consumer Privacy Act which can thus include providing a baseline differential privacy guarantee to all data providers. In the case where some data providers have strong privacy requirements, the constraint $a_i \leq k/m$ can be modified to $a_i \leq k_i/m$ where $k_i$ is chosen appropriately based on the privacy requirements demanded by the data providers. What this does to others' privacy requirements is handled through the IR constraint, i.e., whatever guarantees we provide are such that it is beneficial to all the users in the sense that their payment minus the loss of privacy is guaranteed to be positive. Since this is a model of a market, the assumption is that payment compensates for loss of privacy and so no explicit privacy guarantees need to be calculated.
>
> 2) In our notation, $\eta$ is the privacy budget used for computation and $a_i$ is the weights used for logistic regression. Each user's differential privacy guarantee $\epsilon_i = a_i \eta$. Thus, in our case, $a_i$ is the amount of privacy budget used.
>
> 3) We use the same terminology as Fallah et al, which is the first paper to introduce mechanism design in the scalar estimation context. We feel that changing the terminology will cause confusion to a reader referring to the paper for context. However, we are willing to reconsider this terminology if the reviewer feels strongly about it and has an alternative term for sensitivity.
>
> 4) We would like to make sure that we understand the reviewer's question: we believe an example of the situation that the reviewer mentions is one where there are multiple vital statistics of a single user collected over multiple years. This is an interesting question since that would involve some correlation among the data points. This appears to be out of the scope of the present paper but is an interesting topic for future research. Thanks for the suggestion.
>
> Response to the Broader Impact Concerns:
>
> 1) Although data owners may be willing to trade some privacy for better utility, our framework also allows for additional fixed privacy guarantees that can be to the data providers. This will make sure no one is taken advantage of. In addition, since we explicitly consider generalization error by adding a regularizer on $\boldsymbol{a}$ this ensures that the resulting model provides good results across the spectrum of the population.
>
> 2) Our model framework is valid in scenarios such as healthcare data, where patient information is already within the possession of the hospital. In this context, sellers merely need to grant permission to the hospital (a trusted authority) to utilize their data, specifying their privacy sensitivities in the process. We believe that this is a large application space. We agree that the situation that the reviewer mentions is not handled by our model. However since our intended application space is fairly large, we believe the study of the model suggested by the reviewer would be best handled as a separate problem which would be interesting to study in the future. Thanks for the suggestion.

---

### Review · Reviewer_KyRi · 2024-05-09

**Summary Of Contributions:**

This paper addresses the challenge of solving machine learning tasks with data from privacy-sensitive sellers. The authors designed a mechanism that optimizes the balance between test loss, seller privacy, and payments by first solving logistic regression with differential privacy guarantees, then applying mechanism design theory to handle nonconvex optimization problems.

**Audience:**

Yes

**Broader Impact Concerns:**

No concerns for ethical issues

**Claims And Evidence:**

Yes

**Requested Changes:**

1. How can we ensure that sellers will honestly report their privacy sensitivy? Although authors are talking about IC in equation (6), wouldn't it be possible to report a higher sensitivy in order to receive higher compensation?

2.Is it possible to calculate the total budget consumption based on the target accuracy rather than the accuracy under a limited budget?

**Strengths And Weaknesses:**

Strengths
- Optimizing mechanism design for logistic regression problems
- Changing the non-convex problem of an object function to a convex function

Weaknesses
- Please refer to the requested changes.

---

> ### Author Response · Authors · 2024-05-28
> **Response by the Authors**
>
> We thank the reviewer for their review and their comments. Please find our response to the requested changes below:
>
> 1) How can we ensure that sellers will honestly report their privacy sensitivity? Although the authors are talking about IC in equation (6), wouldn't it be possible to report a higher sensitivity in order to receive higher compensation?
>
> Response: Reporting a higher privacy sensitivity will not lead to a higher payment. If you refer to the proof of theorem 3.2, i.e., on page 18, for given privacy sensitivities $\boldsymbol{c}$, the payments are calculated to be $c_i u(\epsilon_i(c_i)) + \int_{c_i}^{} u(\epsilon_i(z_i)) dz_i$. We can see that the first term is increasing in $c_i$ whereas the second term is a decreasing function of $c_i$. Therefore, reporting a higher privacy sensitivity than the true privacy sensitivity will not increase the payment. Additionally, imposing the IC constraint ensures that the sellers' loss which is the (loss of privacy - payment) is minimal when the reported privacy sensitivity is equal to the true privacy sensitivity. Therefore, this ensures that the sellers are incentivized to report their privacy sensitivities truthfully. For additional details, please refer to the proof of theorem 3.2 on page 18.
>
> 2) Is it possible to calculate the total budget consumption based on the target accuracy rather than the accuracy under a limited budget?
>
> Response: Yes, it is possible to calculate the total budget consumption based on target accuracy. To do that, please refer to Figure 2a. In Figure 2a, if we draw a horizontal line that denotes the target accuracy, the point at which the line meets the blue line will provide the total budget consumption. In other words, we can first start with a higher value of $\gamma$ and then continue to decrease it till we get the required target accuracy.

---

### Review · Reviewer_TpK4 · 2024-05-21

**Summary Of Contributions:**

This paper studies the problem of mechanism design for differentially private logistic regression when the users have variable privacy preferences and need to be compensation for participants is relative to the amount of privacy they are willing to give up (in terms of a larger epsilon). They provide a theoretical foundation for an approach for a (approximately) optimal solution to this problem for minimizing misclassification loss and managing budgetary constraints. Their results show an expected trade-off between misclassification loss of their model and total cost of payments to participants. They show the practicality of their design by showing that the objective function can be made convex.

**Audience:**

Yes

**Broader Impact Concerns:**

I do not have any broader impact concerns.

**Claims And Evidence:**

Yes

**Requested Changes:**

Discuss the privacy implications of the solutions to eqns (3) and (10) being data dependent. If the solution is to use validation data and knowledge of the distribution of privacy preferences then discuss this solution. Joint differential privacy seems like the appropriate model for the privacy of the payments (where payments to user i are differentially private with respect to all user data expect user I, which it can be non-private with respect to). Without this discussion, it seems like the proposed solution has privacy issues.

Minor:
- typo: “The first definition of differential privacy was introduced by Dwork et al. which considered heterogeneous…”
- Typo: “We say that an algorithm provides centrally…”
- In eqn (10), should a_i, nu be a_i x nu?
- Typo: just before Theorem A.1 “Note that”

**Strengths And Weaknesses:**

Strengths: Mechanism design for handling heterogeneous user privacy preferences is an interesting problem that meets a practical need. The techniques employed by the authors to solve this problem are non-trivial and their mechanism seems to perform well on a real world dataset (the Wisconsin breast cancer data set).

I appreciated that the authors discussed how correlation between privacy sensitivities and data points could cause a biased result. This seems to me to be a major issue in this area; as the authors point out, correlation is very much expected, especially in medical datasets. The authors offer some empirical evidence that the regularisation in their mechanism serves to reduce the impact of correlation.

Weaknesses: My main concern is that (as far as I could discern) the authors do not make any claims about the privacy of the resulting model and the payments after solving the objective function? The solution to the optimisation problem in (3) is dependent on both the data and the privacy sensitivities, both of which are sensitive information. How does the resulting logistic regression solution avoid leaking privacy through the weights? The authors mention selecting the hyper-parameters using validation data (which perhaps one could assume was non-private), can the same be done for selecting a weight function? The solution to eqn (10) seems to have the same issue? Do the payments leak information about other users data?

I found the algorithm and theorem statements difficult to follow as they lacked detail and variables often appeared with no explanation. For example, in algorithm 0, many variables are introduced but their role is not properly discussed. In Theorem 2.1, mu and sigma are just defined as “appropriate mu, sigma” and then they are called hyperparameters in eqn (3)? It seems like they depend on beta and delta, which are dropped after Theorem 2.1? They aren’t even introduced in Theorem A.1, although formulas for them are given at the end of the proof.

I would have liked more detail on how the IC and IR properties were enforced? It’s possible this comes from standard mechanism design theory that I am unfamiliar with, but some discussion would be appreciated.

I believe the underlying assumption that the system is given full access to the data and privacy sensitivities before deciding on the payment structure is present in prior work. None-the-less, I thought it warranted justification since this does not seem like the natural setting for such a data auction (where data would not be received until payment is received).

---

> ### Author Response · Authors · 2024-05-28
> **Response by the Authors**
>
> We thank the reviewer for their review and their comments. Please find our response to the weaknesses and the requested changes below:
>
> Response to Weaknesses:
>
> 1) Regarding the first question about not making any claims about the privacy of the resulting model and the second question about the resulting logistic regression solution avoiding leaking privacy through the weights, we note that providing such privacy guarantees is one of the central contributions of this work. For Eq. (3), we choose $(\boldsymbol{a},\eta)$ from the set $\mathbb{F}$. Further, our result in Proposition 1 proves that for any choice of $a,\eta$ in $\mathbb{F}$ the optimal weights $\boldsymbol{w}$ obtained by minimizing $\sum a_i \log(1+e^{-y^i \boldsymbol{w}^T \boldsymbol{x}^i}) + \lambda/2 ||\boldsymbol{w}||^2 + 2 \boldsymbol{b}^T\boldsymbol{w}/\eta$ where $||\boldsymbol{b}||$ is sampled from $\Gamma(n,1)$ with its direction chosen randomly, satisfies $\epsilon_i$ differential privacy guarantees. Therefore, after optimizing Eq. (3), the resulting weights will be released to the public. Now these weights will satisfy $\epsilon_i$ differential privacy guarantees. An intuitive way to understand this is that the term $\boldsymbol{b}^T\boldsymbol{w}/\eta$ adds randomness to the final optimal solution of Eq. (3) which helps avoid leaking privacy when the model weights w are released publicly.
> 2) Regarding selecting weights from validation data, we note that our problem statement does not require a priori guarantees on privacy. We provide privacy guarantees by learning the privacy sensitivities of the sellers, i.e., we pay different amounts to different sellers and provide different privacy guarantees to each seller. One may ask why sellers would agree to this: we prove that sellers would agree to this by establishing that the mechanism is individually rational, i.e., the net result of payment received minus loss of privacy is positive for all sellers. Therefore, we do not need to select weights from validation data.
>
> 3) Regarding the question about payments leaking privacy information, this is a good question. However, we note that the payments will only be released individually to the sellers and not publicly. Therefore, assuming that the data providers/sellers are not adversarial, leakage of information through payments does not risk the data from membership attacks or any other adversarial attacks.
>
> 4) Here's a short explanation of how IC and IR properties are enforced. In mechanism design, we announce a mechanism, i.e., a process for making payments and learning an ML model, and this process takes into account the fact that sellers will not participate in the mechanism if it is not beneficial to them (the IR constraint) and that sellers are incentivized to truthfully reveal their sensitivities (the IC constraint). Mathematically, in the proof of theorem 3.2, one can see that the payments are chosen by enforcing the IC and IR constraints. The proof is provided on page 18.
>
> 5) The mechanism is designed to ensure that it is individually rational, i.e., it benefits the sellers to participate in the mechanism. This is the reason we assume that data is provided before payments.
> Our assumption that the system is given full access to the data is justified in scenarios such as healthcare data, where patient information is already within the possession of the hospital. In this context, sellers merely need to grant permission to the hospital (a trusted authority) to utilize their data, specifying their privacy sensitivities in the process. Further, we also consider that patients can lie about their privacy sensitivities which is then mitigated through designing payments that incentivize the patients to be truthful about their privacy sensitivities.
>
> Response to Requested Changes:
> 1) As mentioned earlier, our mechanism design assures individual rationality, which means that all sellers are happy with their privacy-payment tradeoff, independent of the data. Additionally, as mentioned before, the result of Proposition 1 ensures that the released model weights are consistent with the differential privacy guarantees provided to the data providers. This, along with the payments being individually rational ensures that sellers are benefitted from participating in the mechanism. The reviewer has a valid point about the payments leaking information, but as mentioned before, the payments are not released publicly.
> 2) The reviewer mentions some typos in the paper. Thank you for highlighting them. We will fix that in the revised version.

---

> > ### Comment · Reviewer_TpK4 · 2024-06-10
> > **Question about privacy guarantee**
> >
> > Thank you to the authors for their response.
> >
> > I am still a little confused about the authors response to weakness 1. I understand that for any (a, eta) pair, the algorithm is eps-DP. What's still not clear to me is that if (a, eta) are chosen data dependently, then this still holds true. For example, for binary data the algorithm sum x_i + Lap(b) is eps-DP provided b>1/eps. However, if I say "if person x is in the database then let b=1000/eps, otherwise, let b=1/eps", then this whole mechanism is not eps-DP because I can probably guess the standard deviation and hence learn whether person x was in the database. I don't currently see a reason why the algorithm described does not fall into the trap also?

---

> > > ### Author Response · Authors · 2024-06-11
> > > **The epsilon-DP is wrt to w**
> > >
> > > Thanks for your question. Only the vector $w$ is released publicly. The parameters $a$ and $\eta$ are not released publicly, it is only known to the mechanism designer, i.e., the buyer.  We show that the publicly released vector $w$ has the $\epsilon_i$-DP property for each seller $i.$ We hope this answers your question.

---

> > > > ### Comment · Reviewer_TpK4 · 2024-06-11
> > > > **I'm not sure it matters if you publicly release a and eta or not.**
> > > >
> > > > Thank you for your engagement. My confusion comes from the fact that I'm not convinced that it matters if you publicly release $a$ and $\eta$. In the example I had with the Gaussian mechanism, it doesn't matter if I release sigma or not, the whole mechanism does not satisfy the required privacy guarantee because I can distinguish which mechanism was used *without being told*, violating privacy. Is there a reason that this doesn't happen in this case?

---

> > > > > ### Author Response · Authors · 2024-06-12
> > > > > **Only the weight vector w is released to the public**
> > > > >
> > > > > Thank you for your follow up question. The platform releases the payments and the model weights w. Additionally the payments are only released to the sellers, therefore the only information available to the public (other than the sellers) is the model weights w. Further, the information about how \epsilon is calculated is also not revealed to anyone. Therefore, in your example, the adversary would not know the dependence of b wrt the data and thus cannot make a guess about the data. Also, we prove that for our calculated differential privacy guarantees which determine the payments as a result, the model weights are consistent with the differential privacy guarantees. Thus, the only way through which one could infer anything about the data is through payments. This can be further solved by either a) since the payments are revealed only to the sellers, we assume that the sellers are not adversarial which we believe is a fair assumption b) If we do not assume a) then we can add sufficient amount of zero mean noise to the payments. Further, since the noise is zero mean on average the platform also does not need to pay extra compensation.

---

### Decision · Action_Editor_odxp · 2024-07-21

**Recommendation:** Reject

**Comment:**

The authors formulate an interesting problem for differentially private logistic regression under heterogeneous privacy guarantees using a payment mechanism to trade-off loss and cost. The authors convexify the underlying non-convex optimization problem to solve these trade-offs. The main challenges with the current state of the paper is:
- _clarity of exposition_: The reviewers (and I) found it hard to understand the mathematical contributions, and what was imposed as an assumption, and what guarantees exactly hold. Particularly the IC and IR constraints. Furthermore, the notation was also a bit challenging, where it wasn’t clear what each different component signified.
- _unclear privacy guarantees_: The reviewers were not convinced on the different parameters of the protocol leaking privacy, particularly scale of $b$ depending on $\eta$. Though the authors did try to defend their decisions, both reviewers were not convinced.

Beyond the comments from Reviewer TpK4 in their review, here are some comments from Reviewer rRmq that would be helpful:
  - _Formulations used throughout the paper and the proofs are confusing: while authors claim that the choice of payments make these properties hold, these are always referred to as "using the IC and IR constraints", which does not highlight the fact that these properties arise from the choice of payment._
  - _I disagree with the authors on the fact that their modelization of correlation is "fair", since in 3.5, they merely say that correlation is compensated for by regularization (but the statement remains very vague), and experimental results only provide one specific example of correlations. I find this unsettling since these correlations could give away a lot of private information in the process._
  - _Regarding utility, the authors claim it is difficult to give utility guarantees regarding misclassification rate. This is true, but utility is often measured as error in terms of loss function in the design of DP mechanisms, which should be possible to derive in this case._

As described above, the reviewers felt that the correctness of the paper and validity of the different assumptions of the model remain a bit unclear, and the paper is not ready to be accepted at TMLR. I recommend the authors to improve the exposition of these aspects, add detailed discussion on the different parameters and their dependence on data, add more rigorous and clear theorem statements, and resubmit.

**Audience:**

Yes.

**Claims And Evidence:**

No. The exposition of the paper regarding the IC and IR guarantees are not clear and created confusion among the reviewers. Additionally, privacy leakage through different parameters in the algorithm is not clearly articulated and discussed.

**Resubmission Of Major Revision:**

The authors may consider submitting a major revision at a later time.